# ARID1A loss derepresses a group of human endogenous retrovirus-H loci to modulate BRD4-dependent transcription

Chunhong Yu[1,7], Xiaoyun Lei[1,7], Fang Chen[1], Song Mao[1], Lu Lv[1], Honglu Liu[1], Xueying Hu[1], Runhan Wang[2], Licong Shen[1,3], Na Zhang[1], Yang Meng[2], Yunfan Shen[2], Jiale Chen [2], Pishun Li[1], Shi Huang [2], Changwei Lin [1,4], Zhuohua Zhang[1,2,5] & Kai Yuan [1,2,5,6 ✉]

Transposable elements (TEs) through evolutionary exaptation have become an integral part of the human genome, offering ample regulatory sequences and shaping chromatin 3D architecture. While the functional impacts of TE-derived sequences on early embryogenesis have been recognized, their roles in malignancy are only starting to emerge. Here we show that many TEs, especially the pluripotency-related human endogenous retrovirus H (HERVH), are abnormally activated in colorectal cancer (CRC) samples. Transcriptional upregulation of HERVH is associated with mutations of several tumor suppressors, particularly ARID1A. Knockout of ARID1A in CRC cells leads to increased transcription at several *HERVH* loci, which involves compensatory contribution by ARID1B. Suppression of HERVH in CRC cells and patient-derived organoids impairs tumor growth. Mechanistically, HERVH transcripts colocalize with nuclear BRD4 foci, modulating their dynamics and co-regulating many target genes. Altogether, we uncover a critical role for ARID1A in restraining HERVH, whose abnormal activation can promote tumorigenesis by stimulating BRD4-dependent transcription.

[1] Hunan Key Laboratory of Molecular Precision Medicine, Department of Oncology, Xiangya Hospital, Central South University, Changsha, Hunan, China. [2] Hunan Key Laboratory of Medical Genetics, School of Life Sciences, Central South University, Changsha, Hunan, China. [3] Department of Gynecology, Xiangya Hospital, Central South University, Changsha, Hunan, China. [4] Department of Gastrointestinal Surgery, The Third Xiangya Hospital, Central South University, Changsha, Hunan, China. [5] National Clinical Research Center for Geriatric Disorders, Xiangya Hospital, Central South University, Changsha, Hunan, China. [6] The Biobank of Xiangya Hospital, Central South University, Changsha, Hunan, China. [7] These authors contributed equally: Chunhong Yu, Xiaoyun Lei. ✉ email: yuankai@csu.edu.cn

We have been facing constant viral attacks during the course of evolution. While most viruses come and go, few have invaded and colonized the germline genome, becoming a significant fraction of transposable elements (TEs) that contribute more than 50% to the human nuclear DNA content[1–4]. Human TEs include DNA transposons, long terminal repeat (LTR) retrotransposons, and non-LTR retrotransposons. The majority of them have lost the ability to transpose during evolution and had long been regarded as functionless repetitive DNA. Recent studies however have begun to reveal that TEs are an abundant source of many regulatory sequences[2,3,5,6], such as microRNAs (miRNAs) and long noncoding RNAs (lncRNAs)[7–12], and that TEs are co-opted to serve important functions including transcriptional regulation, chromatin organization, and 3D compartmentalization, especially during early embryogenesis and in embryonic stem cells (ESCs)[6,13–23].

The endogenous retroviruses (ERVs), which have been identified half a century ago[24], make up 8% of the human genome. They are LTR retrotransposons and have similar compositions to retroviruses, with internal coding sequences (gag-pro-pol-env) flanked by a pair of identical LTRs containing cis-regulatory elements. By estimation, there are 98,000 copies of ERVs and their derivatives in the human genome, with human endogenous retrovirus H (HERVH) being one of the most abundant groups. Like other human ERVs, most of the HERVH elements are no longer intact but truncated forms and solitary LTRs, and only approximately 100 copies are close to full-length[25,26]. Human ERVs are largely in heterochromatin and transcriptionally repressed by an expanding battery of epigenetic mechanisms[4,27,28], including methylation of histone H3 on lysine 9 (H3K9) or lysine 27 (H3K27), DNA methylation, as well as the RNA N(6)-methyladenosine (m(6)A) modification[29,30]. Of note, these regulatory mechanisms are often redundant and function in a context-specific manner[31,32], reflecting the sophisticated evolutionary arms race between viral sequences and the host genome[2–4].

ERVs in the human genome are not always inactive. During the profound epigenetic resetting in early embryonic development, ERVs are systematically transcribed in a stage-specific manner, coinciding with different cellular identities and differentiation potencies[33]. While a comprehensive understanding of ERVs' functions during early embryogenesis is yet to be established, recent studies have revealed the intimate relationship between HERVH and the human pluripotency network[19]. Depending on different variants of LTR (LTR7, LTR7Y, and LTR7A/B/C), the transcription of HERVH internal sequence (HERVH-int) is activated from 4-cell stage to blastocyst[33]. HERVH transcripts are also highly abundant in human ESCs as well as induced pluripotent stem cells (iPSCs), and moreover, the naïve-like pluripotency is associated with higher levels of HERVH expression[15,16,19,26]. Activation of HERVH promotes both the acquisition and maintenance of pluripotent states, by generating noncoding RNAs (ncRNAs) or producing chimeric transcripts with protein-coding genes via alternative splicing[15,16]. The transcriptionally active HERVH can also demarcate topologically associated domains (TADs) and help maintain a pluripotent chromatin architecture[22]. Cancer development in many aspects parallels the process of early embryogenesis. This includes regaining the capacity of self-renewal and dramatic alterations in epigenetic landscapes. Interestingly, reactivation of HERVH is also observed in several types of human cancer, such as colorectal carcinomas (CRCs)[34–39], however, a mechanistic insight into this reactivation is lacking and its functional consequence unclear.

The SWI/SNF (mating type SWItch/Sucrose NonFermentable) family chromatin remodelers, BAF, PBAF, and GBAF, regulate chromatin packing and transcription by controlling the dynamics of nucleosomes[40]. As a subunit of the BAF complex, ARID1A functions as a bona fide tumor suppressor and is mutated in approximately 8% of all human cancers[40–44]. Mutation of ARID1A sensitizes cancer cells to bromodomain and extra-terminal domain (BET) inhibitors[45,46], likely due to its indispensable role in maintaining normal enhancer function by influencing BRD4 activity[42,46,47]. How ARID1A mutation affects BRD4 remains unknown. Here, we show that loss of ARID1A results in the derepression of several HERVH loci, and the transcribed HERVH RNA partitions into nuclear BRD4 puncta and contributes to the BRD4-dependent gene regulatory network. This HERVH–BRD4 axis is crucial for the growth of CRC cells and patient-derived organoids, offering potential treatment opportunities for the ARID1A mutated colorectal cancers.

## Results

**The expression of HERVH is abnormally upregulated in CRCs.** The majority of the human genome is comprised of various repetitive DNA sequences, most of which are transcribable. To globally characterize the expression of repetitive DNA elements in CRCs, we collected 521 colon adenocarcinoma (COAD) and 177 rectum adenocarcinoma (READ) RNA-seq data from The Cancer Genome Atlas (TCGA), filtered and grouped them according to the variables (Supplementary Fig. 1a), and quantified the repeats expression using the human RepeatMasker Repeats annotation (https://genome.ucsc.edu/cgi-bin/hgTables). We first applied principal component analysis (PCA) to the gene expression as well as the repeats expression data from 51 normal and 631 tumor samples (Supplementary Fig. 1a). Both the genes and repeats showed distinct expression profiles that successfully demarcated the normal and tumor samples (Fig. 1a, b). We next categorized the differentially expressed repeats. While the simple repeats were the most abundant, many LTR retrotransposons (also known as ERVs) showed altered expression between normal and tumor tissues (Fig. 1c). Of the 580 ERVs, 44 were downregulated and 84 upregulated in CRC tumors (Fig. 1d). To validate these upregulated ERVs, we repeated the analysis with another independent RNA-seq dataset (GSE50760) of CRC samples and identified 17 ERVs that showed consistent upregulation. Specific activation of ERVs is linked with pluripotency in embryonic cells[6,20]. We compared the upregulated ERVs in CRC tissues with that observed in early embryos and ESCs and pinpointed two elements, HERVH-int and LTR7Y, that constitute a full-length HERVH (Fig. 1e). Both elements showed increased expression in tumor tissues and their expressions were highly correlated with each other in the TCGA colorectal dataset (COREAD) (Supplementary Fig. 1b–d and Supplementary Data 1). To further confirm the upregulation of HERVH in CRCs, we performed RNAscope analysis for HERVH RNA on CRC tissue array. Compared with the matched peritumoral tissues, the tumoral tissues showed significantly stronger RNAscope signals (Supplementary Fig. 1e, f). We then investigated the association of the expression levels of HERVH with the clinical outcomes of the CRC patients using the TCGA COREAD dataset, and observed that higher HERVH-int expression predicted poorer survival (Fig. 1f).

Molecular characterization of CRC samples has revealed 24 genes that are significantly mutated[48]. To interrogate the relationship between these gene mutations and the upregulation of HERVH, we selected 516 CRC samples with genetic variation data from the TCGA COREAD dataset, extracted their mutational signatures, and correlated the mutational status of each one of the 24 genes with the expressions of HERVH-int and HERVK-int for comparison (Supplementary Fig. 1a). In contrast to HERVK-int whose expression showed no obvious association with any gene mutations analyzed, the expression of HERVH-int correlated with the mutational status of several genes (Fig. 1g).

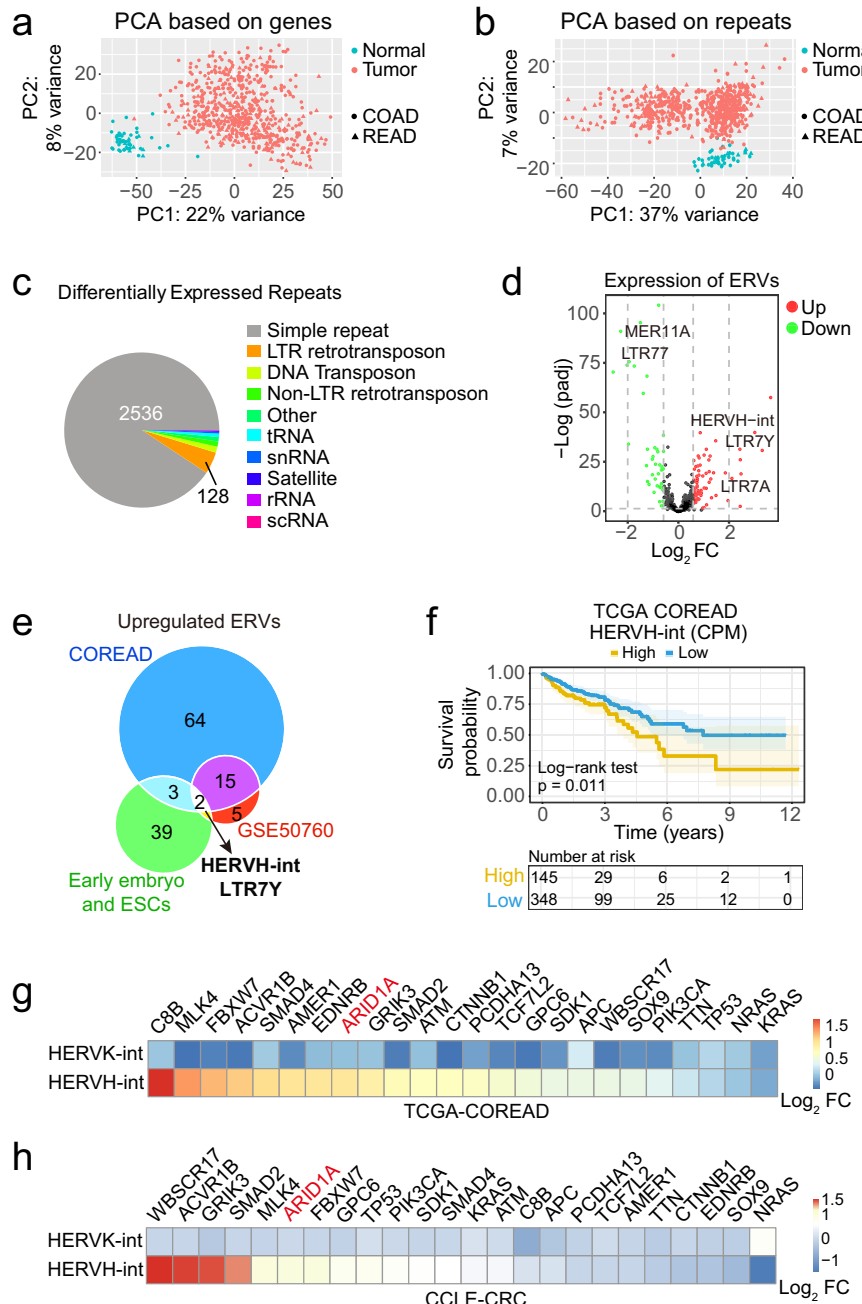

**Fig. 1 Characterization of HERVH expression in CRCs. a** PCA of gene expression of 51 normal and 631 CRC tumor tissues from the TCGA COREAD dataset. **b** PCA based on the expression of repetitive sequences. **c** Classification of differentially expressed repetitive sequences analyzed using DESeq2 (version v1.22.2, with two-tailed likelihood ratio test). Differentially expressed repetitive elements are determined with the cut-off values of adjusted $p$ value <0.05 and |Log$_2$ FC (Fold Change)| >0.585. **d** Volcano plot of the differentially expressed ERVs analyzed using DESeq2 (version v1.22.2, with two-tailed likelihood ratio test). Up (red) and down (green) regulated ERVs are determined with the cut-off values of adjusted $p$ value <0.05 and |Log$_2$ FC | > 0.585. **e** Overlap analysis of the upregulated ERVs in CRCs samples and early embryonic cells identifies the internal coding sequences of HERVH (HERVH-int) and its corresponding LTR (LTR7Y) as the commonly upregulated elements. **f** Survival analysis based on the expression level of HERVH-int and the overall survival (OS) from 493 patients with AJCC pathologic tumor stage >I. The mean expression value of HERVH-int is used to demarcate the HERVH-int-High (145 patients) and HERVH-int-Low (348 patients) groups. Log-rank test, $p = 0.011$. **g** Correlation analysis of HERVH-int expression and mutational status of the most frequently mutated genes in CRCs using the TCGA dataset. **h** Correlation of HERVH-int expression and gene mutations in CRC cell lines in the CCLE dataset. The data used to generate each panel in Fig. 1 are listed in Supplementary Data 1. Source data including exact $p$ values are provided as a Source data file.

We expanded this analysis to the 59 CRC cell lines in Cancer Cell Line Encyclopedia (CCLE)[49] (Fig. 1h), and identified a list of genes whose mutation was consistently correlated with upregulation of HERVH. This included MLK4, FBXW7, ACVR1B, ARID1A, GRIK3, and SMAD2.

**Loss of ARID1A leads to transcriptional activation of HERVH.** Knockdown (KD) the expression of some of the listed genes by small interfering RNA (siRNA) resulted in increased transcription of HERVH (Supplementary Fig. 1g). We selected ARID1A for further functional validation because it is a core

subunit of the BAF chromatin remodeler complex and its inactivation mutations occur in a broad spectrum of human cancers[40,41,43].

To comprehensively depict the changes of repeats expression upon ARID1A loss, we collected and analyzed two independent RNA-seq data of HCT116 wild type (WT) and its isogenic ARID1A knockout (KO) cell lines[42,50] (Supplementary Fig. 2a). Of note, the LTR retrotransposons or ERVs were the most upregulated repeat group in ARID1A KO cells (Fig. 2a and Supplementary Data 2). We ranked all the ERVs according to their fold changes (Fig. 2b and Supplementary Fig. 2c) and found that HERVH-int and two of its associated LTRs, LTR7Y, and LTR7, were the three most significantly upregulated elements, whereas other HERVH-related LTRs did not show consistent upregulation (Fig. 2c). Scatter plots of the expression of all 580 ERVs revealed that HERVH-int, LTR7Y, and LTR7 were already expressed in HCT116 WT cells but the ARID1A inactivation further increased their abundance (Fig. 2d and Supplementary Fig. 2d). These observations were further validated with our own RNA-seq data of ARID1A WT and KO HCT116 cells (Supplementary Fig. 2e, f and Supplementary Data 2). Overlapping the significantly upregulated ERVs in the three datasets spotted HERVH-int and its LTR as the only unambiguously activated elements upon ARID1A loss (Fig. 2e). We generated additional ARID1A KO colorectal cell lines to further confirm the observed upregulation of HERVH (Fig. 2f and Supplementary Fig. 2b). qPCR analyses with primers specifically targeting the *gag* and *pol* sequences of HERVH-int revealed increased transcripts abundance in all three ARID1A KO cell lines (Fig. 2g, h). To test if the transcriptional activation of HERVH can be suppressed by re-expression of ARID1A, we infected the ARID1A KO cells with lentiviruses carrying ARID1A or its counterpart ARID1B[51,52] (Fig. 2i–k). The re-introduction of ARID1A significantly downregulated the expression of HERVH (Fig. 2j). Interestingly, overexpression of ARID1B didn't rescue but instead mildly increased the amount of HERVH transcripts in ARID1A KO cells (Fig. 2k).

Unlike genes, ERVs are quite diverse between primates and rodents. To assess the effect of ARID1A inactivation on ERVs expression in mice, we analyzed RNA-seq data of the colon epithelial cells from WT or ARID1A KO mice[42]. Of the 423 ERVs in the mouse genome, 16 were downregulated and 11 upregulated in the absence of ARID1A (Supplementary Fig. 2g and Supplementary Data 2). The upregulated ERVs included RLTR1B-int, RLTR1D, and RMER12B (Supplementary Fig. 2h). Therefore, the influence of ARID1A on ERVs seemed to be universal, and in human, the most responsive element toward ARID1A mutation was the HERVH.

**ARID1B contributes to the activation of HERVH in the absence of ARID1A.** To investigate how ARID1A loss induced HERVH transcription, we turned our focus to ARID1B, which shares 60% homology with ARID1A[51]. Both ARID1A and ARID1B can function as the rigid structural core in the BAF complex, and they are mutually exclusive[52,53]. ARID1B is essential for the survival of ARID1A mutated cancer cells, by supplying residual BAF complex activities to maintain chromatin accessibility at enhancers and regulate RNA polymerase II dynamics[50,54,55].

We first examined the influence of ARID1B on the expression of repetitive elements using published RNA-seq data[50] (Fig. 3a and Supplementary Data 3). Knocking down the expression of ARID1B in WT HCT116 cells (WT-KD) showed limited effects on repeats expression, however, ARID1B KD in ARID1A KO cells (KO-KD) dramatically reduced the transcripts abundance of many repeats, especially the LTR retrotransposons (ERVs) (Fig. 3b). Particularly, the upregulation of several HERVH elements was markedly reversed by ARID1B KD (Fig. 3c). We confirmed this impact of ARID1B KD on the expression of HERVH in ARID1A KO cells using two different short hairpin RNAs (shRNAs) targeting ARID1B (Fig. 3d). Moreover, we found that re-introduction of a mutant ARID1B insensitive to shRNA restored the high expression level of HERVH in the treated ARID1A KO cells (Fig. 3e).

HERVH as a metagene refers to a group of evolutionarily related genomic elements, including the HERVH-int and different LTR variants (Fig. 2c). Despite the overall sequence similarities, there are significant nucleotide differences among some of these elements that can be used to uniquely map their genomic positions. To identify the specific *HERVH* loci activated by loss of ARID1A, we took in only the uniquely mapped reads and pinpointed 27 *HERVH* elements that were unambiguously derepressed in ARID1A KO cells. 25 of the 27 activated *HERVH* elements showed reduced expression upon ARID1B KD (Fig. 3f and Supplementary Data 3). We further analyzed the changes in chromatin accessibility and several epigenetic modifications on these 25 *HERVH* loci using the published dataset GSE101966[50]. The chromatin accessibility remained unaltered at the derepressed *HERVH* loci, with only a slight increase downstream of these elements (Supplementary Fig. 3a and Supplementary Data 4). Interestingly, acetylation of H3K27 (H3K27ac), as well as mono methylation of histone 3 on lysine 4 (H3K4me), was markedly increased at the 25 activated *HERVH* loci compared with the *HERVH* elements that remained repressed (Supplementary Fig. 3b, c). Of note, although ARID1B KD significantly reduced the HERVH expression, it did not influence the epigenetic signatures analyzed here. One possible explanation was that ARID1B was essential for the viability of ARID1A mutated cells, which limited its KD efficiency (65% KD efficiency in the analyzed dataset).

Given a full-length HERVH comprises HERVH-int and the flanking LTRs, the interspersed HERVH elements in close proximity can be merged into bigger clusters[16]. Accordingly, we merged the activated *HERVH* loci into 13 *HERVH* clusters, two of which had LTRs at both ends (Fig. 3f, *HERVH_3* and *HERVH_9*). We selected the *HERVH_3* cluster for further characterization because it was most highly activated in ARID1A KO cells (Fig. 3g). Using two different primer sets specifically targeting *HERVH_3*, we analyzed the changes in occupancy of ARID1A and ARID1B by ChIP-qPCR. As the amount of ARID1A on *HERVH_3* was reduced in ARID1A KO cells, the binding of ARID1B to this region was increased compensatorily, maintaining a comparable amount of BAF activity on *HERVH_3* (Fig. 3h–j). We also observed an increase in H3K27ac at the *HERVH_3* cluster, consistent with the results generated from the published dataset (Supplementary Fig. 3d–f).

Gain of SP1 sites is associated with transcriptional activation of *HERVH* loci[56]. To verify if SP1 was involved in the activation of the subset of *HERVH* elements in ARID1A mutated HCT116 cells, we knocked down its expression by siRNA and performed RNA-seq analysis (Supplementary Fig. 3g and Supplementary Data 4). Two additional transcription factors (TF) that were predicted to bind *HERVH* were included for comparison[11]. Among the three TFs analyzed, only SP1 knockdown significantly reduced the expression of HERVH (Supplementary Fig. 3f, g). We further compared the density of SP1 motifs between the derepressed *HERVH* group and that remained silenced, and we found that the former harbored more SP1 binding motifs (Supplementary Fig. 3h and Supplementary Data 4), suggesting that SP1 was contributing to the locus-specific activation of HERVH upon ARID1A loss.

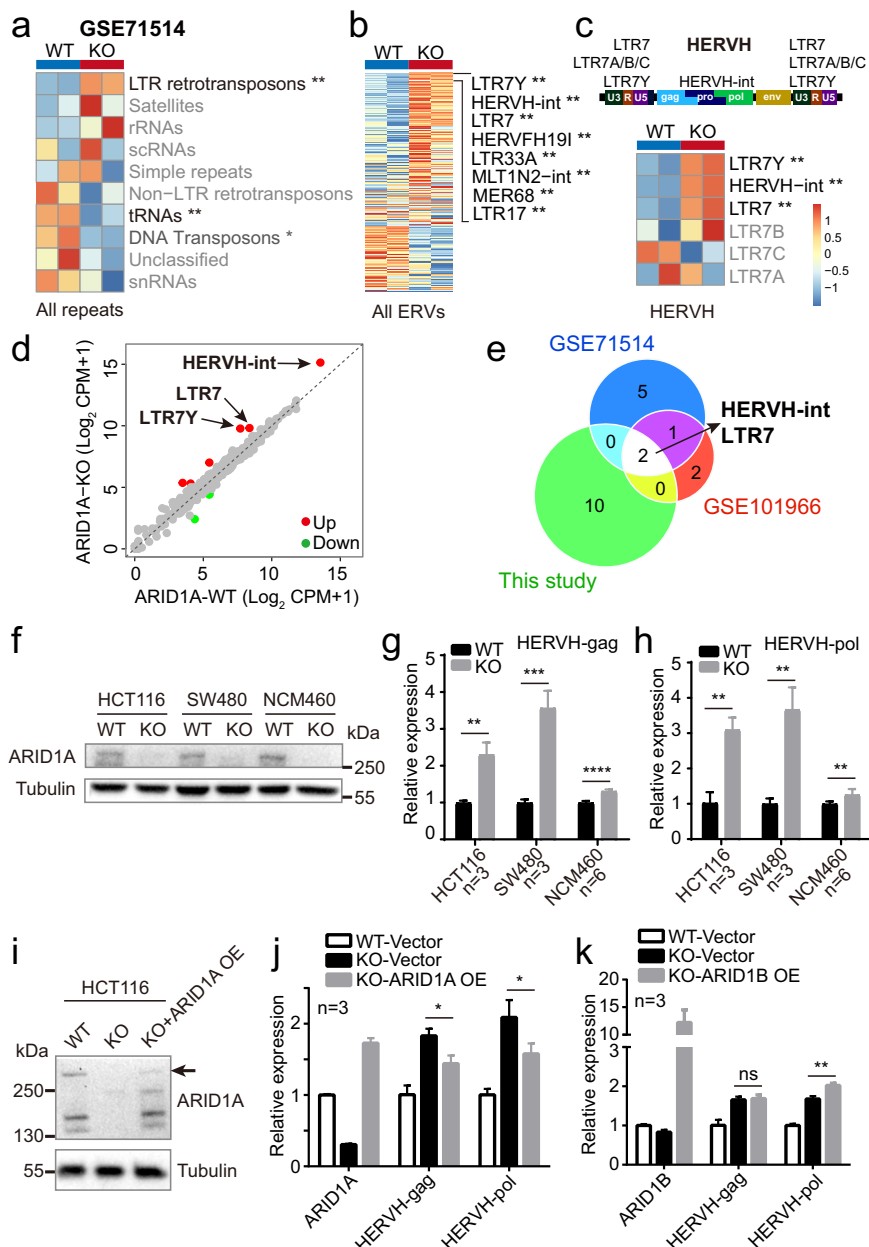

**Fig. 2 ARID1A loss leads to upregulation of HERVH. a–c** Heatmaps of the expression of different repetitive sequences in wild-type (WT) and ARID1A knockout (KO) HCT116 cells. The differential expression is tested based on a model using the negative binomial distribution by DESeq2 (version v1.22.2, with two-tailed likelihood ratio test). The adjusted $p$ values are labeled as $*p < 0.05$, $**p < 0.01$. The schematic of a typical full-length HERVH element is shown in **c**. **d** Scatter plot of the expression of all 580 ERVs in WT and KO HCT116 cells. The up (red) and down (green) regulated ERVs are determined by DESeq2 (version v1.22.2, with two-tailed likelihood ratio test) using cut-off values of adjusted $p$ value <0.05 and $|Log_2 FC| > 1$. **e** Venn diagram showing that HERVH is repetitively upregulated in three independent sequencing experiments with HCT116 ARID1A WT and KO cells. **f** Western blots of different ARID1A WT and KO cell lines. Results are representative of three independent experiments. **g, h** qPCR analyses of HERVH expression in ARID1A WT and KO cell lines using two different primer sets. Data are presented as mean values ± SD from at least three independent experiments, two-tailed unpaired $t$ test, $**p < 0.01$, $***p < 0.001$, $****p < 0.0001$. **i** Western blots showing ARID1A protein levels in WT, KO, and ARID1A rescued KO HCT116 cells. Results are representative of three independent experiments. Arrow indicates the full-length ARID1A band. **j** qPCR analysis of ARID1A and HERVH expression in the indicated cells. Data are presented as mean values ± SD from three independent experiments, two-tailed unpaired $t$ test, $*p < 0.05$. **k** qPCR analysis of ARID1B and HERVH expression. Data are presented as mean values ± SD from three independent experiments, two-tailed unpaired $t$ test, ns: not significant, $**p < 0.01$. Source data including exact $p$-values are provided as a Source data file.

**Expression of HERVH is required for the survival of colorectal cancer cells.** Knockdown the expression of HERVH in ESCs triggers differentiation[15]. To investigate the function of HERVH transcription in CRCs, we knocked down its expression in cell lines and patient-derived organoids using three different shRNAs targeting the HERVH-int region and assessed the consequences

(Fig. 4a). Of note, given that the HERVH transcripts were produced from multiple evolutionarily related genomic loci, different shRNAs demonstrated certain selectivity toward different subsets of the *HERVH* elements (Supplementary Data 7).

We first evaluated the effects of HERVH knockdown in WT and ARID1A KO HCT116 cells. All the three shRNAs treatments

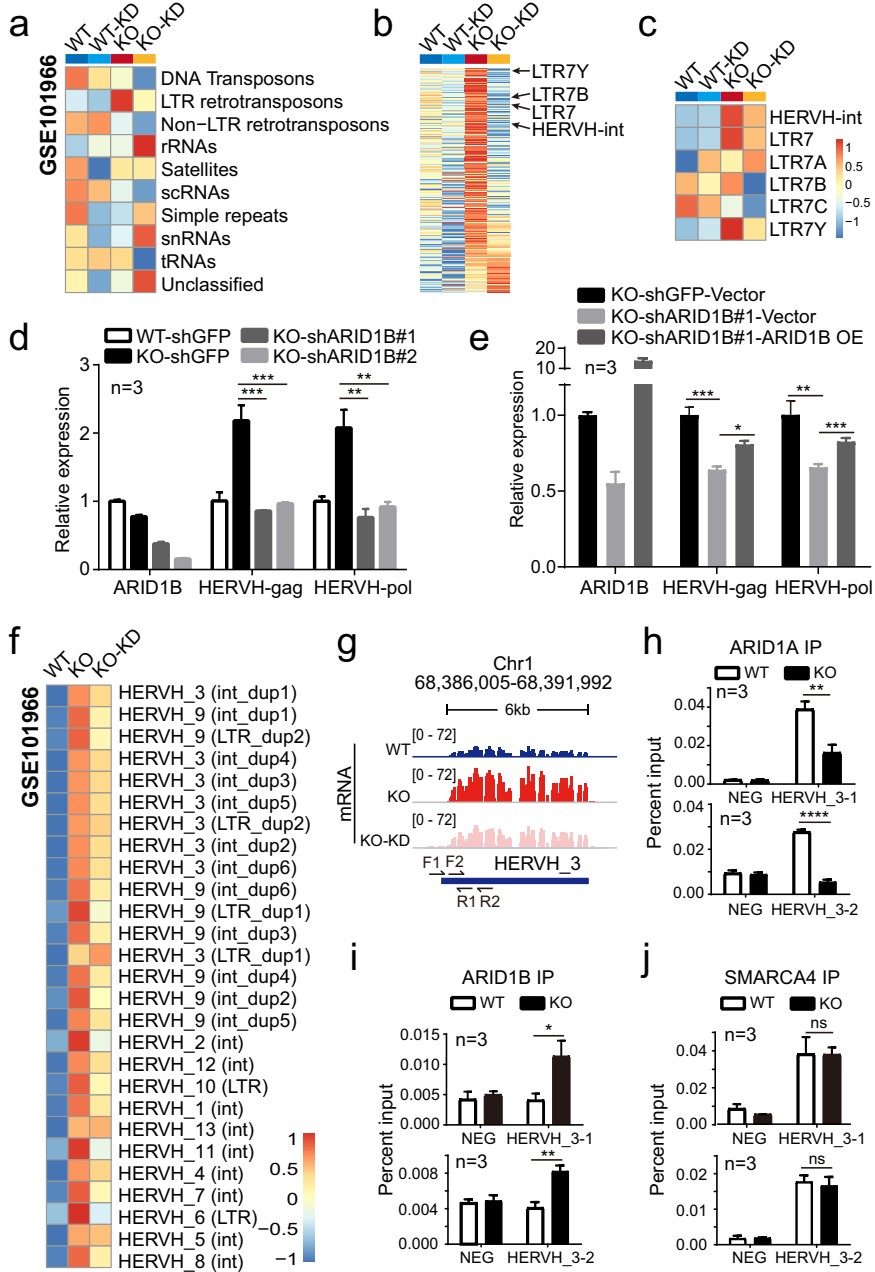

**Fig. 3 ARID1B contributes to the upregulation of HERVH in the absence of ARID1A. a–c** Heatmaps of the expression levels of different repetitive sequences in WT or ARID1A KO HCT116 cells treated with control or ARID1B (KD) shRNA generated with the GSE101966 dataset. **d** qPCR results showing the expression levels of ARID1B and HERVH in ARID1A WT or KO cells treated with shRNA targeting GFP or ARID1B. Two different shRNA sequences targeting ARID1B are used to exclude potential off-target effects. Data are presented as mean values ± SD from three independent experiments, two-tailed unpaired *t* test, **p < 0.01, ***p < 0.001. **e** qPCR analysis of ARID1B and HERVH expression levels in ARID1A KO cells treated with the indicated shRNAs and overexpression plasmids. Data are presented as mean values ± SD from three independent experiments, two-tailed unpaired *t* test, *p < 0.05, **p < 0.01, ***p < 0.001. **f** Heatmap showing the expression levels of the derepressed *HERVH* loci (27) in ARID1A KO cells determined by DESeq2 (version v1.22.2, with two-tailed likelihood ratio test, using cut-off values of adjusted *p* value <0.05 and |Log2 FC | > 0.585). The majority of them (25) display reduced expression when ARID1B is knocked down (KO-KD). **g** Genomic snapshot of RNA-seq signals at a representative *HERVH* locus (*HERVH_3*). Two primer sets targeting this locus for ChIP-qPCR analysis are shown. **h–j** ChIP-qPCR results with two different primer sets showing decreased ARID1A and increased ARID1B at the *HERVH_3* locus in ARID1A KO cells, whereas the amount of another BAF component SMARCA4 remains unchanged. Data are presented as mean values ± SD from three independent experiments, two-tailed unpaired *t* test, ns: not significant, *p < 0.05, **p < 0.01, ****p < 0.0001. Source data including exact *p* values are provided as a Source data file.

resulted in consistent impairment of cell viability and reduction of colony formation (Fig. 4b, c). We expanded the analysis to additional 10 cell lines of colorectal origin. Cell viability assays showed that knockdown of HERVH impaired the survival of most of the cells tested, except the FHC (fetal human cells) cell line established from normal fetal colonic mucosa, which manifested reduced sensitivity (Supplementary Fig. 4a). To rule out possible off-target effect, we treated mice embryonic stem cells (E14) and melanoma cells (B16F10) with HERVH shRNA. HERVH is not conserved and does not exist in the mice genome,

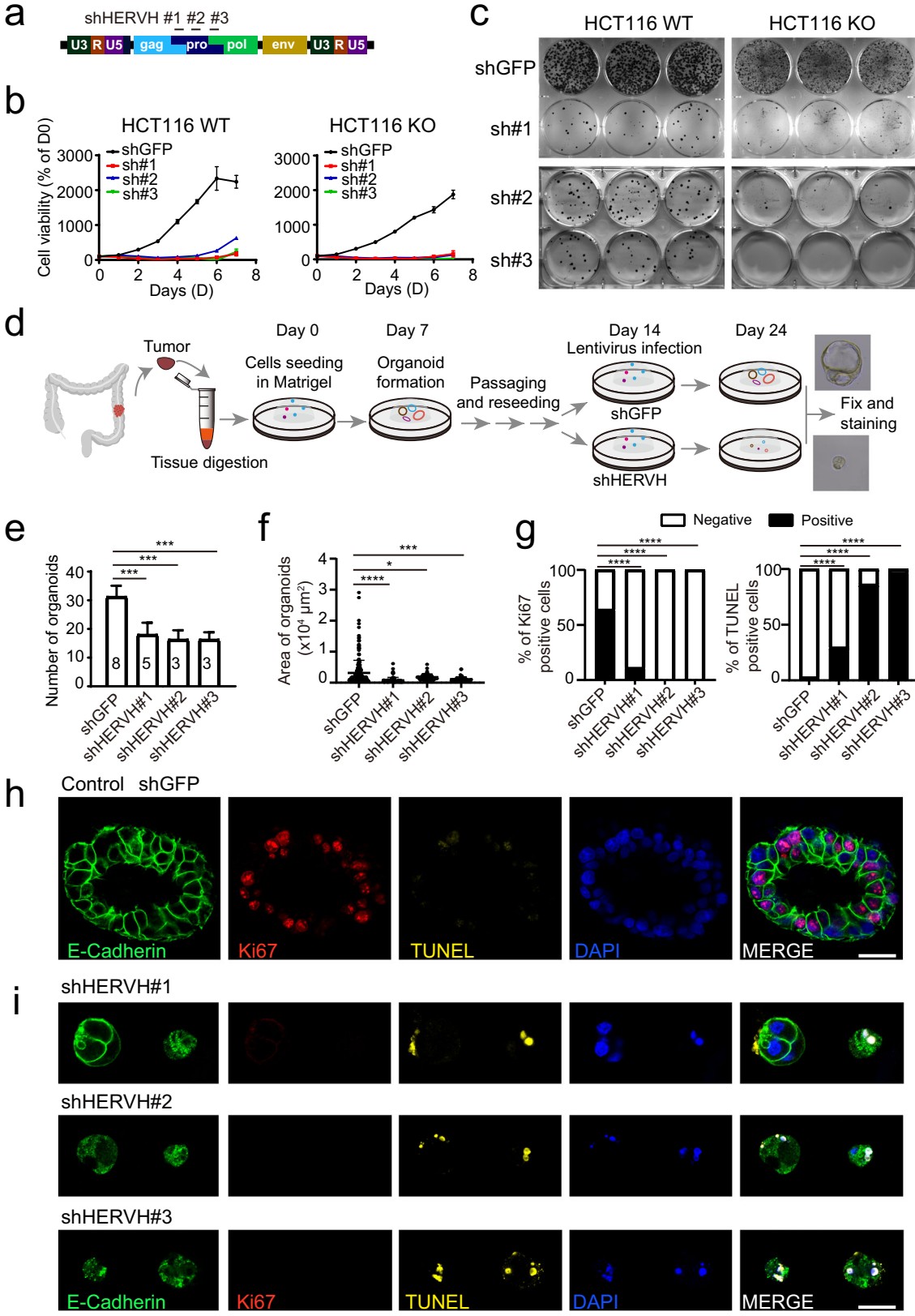

and consistently, we observed no growth inhibition with these cells (Supplementary Fig. 4b). Additionally, the cervical cancer cell line HELA and the osteosarcoma cell line U2OS also showed minimum sensitivity toward HERVH knockdown (Supplementary Fig. 4c), suggesting the existence of tissue specificity that needed future investigation. We next characterized the impact of

HERVH knockdown on the formation of tumor spheres of CRC cells. The CRC cell line SW480 showed a weak ability to form tumor spheres when cultured in 3D Matrigel on ultra-low attachment plates, and loss of ARID1A significantly enhanced this ability (Supplementary Fig. 4d). Knockdown of HERVH in the ARID1A KO cells greatly reduced both the number and the

**Fig. 4 HERVH is essential for the proliferation of CRC cells. a** Schematic showing a full-length HERVH element and the regions targeted by three different shRNAs. **b** The viability of WT and ARID1A KO HCT116 cells after treatment with shRNA targeting GFP control or HERVH. Three different HERVH shRNAs are used to avoid the possible off-target effects. Data presented are from at least three independent experiments. **c** Clonogenic assay showing reduced colony formation in the indicated cells treated with different HERVH shRNAs. Results are representative of three independent experiments. **d** Schematic illustrating the establishment and subsequent treatments of the patient-derived CRC organoids. **e, f** HERVH knockdowns decrease both the number and the size of CRC organoids. Data are presented as mean values ± SD from at least three independent experiments, two-tailed unpaired t test, *p < 0.05, ***p < 0.001, ****p < 0.0001. **g** Percentages of Ki67- or TUNEL-positive cells in control and HERVH shRNA treated CRC organoids. Data presented are from three independent experiments. ****p < 0.0001 by chi-squared test. **h, i** Representative images of control and HERVH shRNA treated organoids stained with E-Cadherin (green), Ki67 (red), TUNEL (yellow), and DAPI (blue). Results are representative of three independent experiments. Bars: 34 µm. Source data including exact n numbers and p values are provided as a Source data file.

size of the observed tumor spheres (Supplementary Fig. 4e, f). We also assessed the contribution of HERVH to the tumorigenicity of CRC cells. We treated the ARID1A WT or KO HCT116 cells with control or shRNA targeting HERVH and subcutaneously seeded them into nude mice. The cells with HERVH knocked down formed much smaller tumors on the 12th day after inoculation (Supplementary Fig. 4g).

To further verify the critical role of HERVH for CRCs, we obtained tumoral and peritumoral tissues from surgical biopsy, evaluated their HERVH transcripts levels (Supplementary Fig. 4h), and established patient-derived CRC organoids (Fig. 4d). The CRC organoids were infected with lentiviruses carrying shRNAs targeting either HERVH or GFP to achieve specific knockdown (Supplementary Fig. 4i). Compared to control, HERVH knockdowns resulted in the formation of fewer organoids (Fig. 4e), and their sizes were much smaller as well (Fig. 4f). We examined the cell proliferation and apoptosis in the treated organoids by Ki67 and TUNEL stainings (Fig. 4g–i). HERVH knockdown dramatically reduced the number of proliferating Ki67 positive cells, meanwhile increased the number of TUNEL positive apoptotic cells (Fig. 4g). Altogether, these results suggested that HERVH was a vulnerability of colorectal cancer cells.

**HERVH transcripts contribute to the BRD4-dependent transcriptional network.** HERVH RNA is part of the transcriptional circuitry regulating pluripotency[13,15,16,57]. To investigate the molecular underpinnings of the oncogenic function of HERVH in CRCs, we assessed the impact of HERVH knockdown on global gene expression. PCA showed that the transcriptomes of the ARID1A WT and KO HCT116 cells were noticeably separated on the second principal component (PC2), and HERVH knockdown narrowed this difference (Fig. 5a), suggesting that the altered transcription seen in ARID1A KO cells was partially linked to the activation of HERVH. We identified 552 upregulated genes and 531 downregulated genes whose transcriptional changes were reversed upon HERVH knockdown in ARID1A KO cells (Fig. 5b and Supplementary Data 5). Many of the 552 HERVH-dependent upregulated genes in ARID1A KO cells were enriched in cancer-related pathways (Fig. 5c). We selected some representative target genes and validated their HERVH-dependency by qPCR (Fig. 5d, e).

HERVH RNA is able to interact with many subunits of the mediator complex[15], and the observed increase in H3K27ac and H3K4me at the derepressed HERVH loci suggested that they could function as active enhancers and their transcripts enhancer RNAs[58]. We compared the transcriptome dynamics after siRNA-mediated knockdown of individual subunits of the mediator complex, its binding partner BRD4, and HERVH (Supplementary Fig. 5a). The correlation matrix suggested that suppression of BRD4 and HERVH imposed similar influences on global transcription when compared with other mediator components (Fig. 5f). Further analysis using the differentially expressed genes

in BRD4 and HERVH knockdown cells revealed an even stronger correlation between these two groups (Fig. 5g and Supplementary Data 5). Of the 1643 differentially expressed genes upon HERVH knockdown, 1018 of them showed similar changes in BRD4 knockdown cells (Fig. 5h and Supplementary Fig. 5b).

To localize the HERVH RNA inside the cell, we designed and prepared fluorescence in situ hybridization (FISH) probes targeting the *gag* sequence of HERVH-int (Supplementary Data 6). The specificity of the FISH probe was validated by the much-reduced fluorescent signals in the ARID1A KO HCT116 cells after HERVH knockdown (Supplementary Fig. 5c). We combined RNA FISH targeting HERVH with immunofluorescence (IF) staining to interrogate the subcellular localizations of HERVH transcripts and components in the mediator coactivator complex. Varying degrees of colocalization were detected between HERVH transcripts and the endogenous BRD4, MED1, and MED12 (Supplementary Fig. 5d), suggesting that HERVH RNA could regulate their protein dynamics in the nucleus.

**HERVH RNA regulates BRD4 dynamics in the nucleus.** BRD4 as well as the mediator complex subunit MED1 can form liquid-like condensates, especially at super-enhancers[59,60]. We stably expressed GFP-BRD4 in the ARID1A KO HCT116 cells, and observed nuclear condensates as previously reported[59]. Moreover, RNA FISH revealed a clear distribution of HERVH transcripts in these GFP-BRD4 puncta (Fig. 5i).

To investigate the contribution of HERVH RNA in the formation and dynamics of the BRD4 puncta, we knocked down the expression of HERVH in the ARID1A KO HCT116 cells stably expressing GFP-BRD4. Downregulation of HERVH transcripts abundance with two different shRNAs both resulted in a marked decrease in the number as well as the size of the BRD4 puncta (Fig. 6a). Similar results were observed in the cells treated with shRNAs targeting SP1, validating its function as the main TF in HERVH transcription (Supplementary Fig. 6a). We further assessed the dynamics of the BRD4 puncta using fluorescence recovery after photobleaching (FRAP). While in the control cells the photobleached BRD4 punctum quickly restored its fluorescence, the fluorescence recovery after HERVH knockdown became much slower (Fig. 6b). The BRD4 protein level also manifested a mild decrease when HERVH was knocked down (Fig. 6c).

We wanted to estimate the contribution of the compromised BRD4 dynamics to the observed impairment of survival of CRC cells after HERVH knockdown. 1,6-hexanediol (1,6-HD) is widely used to dissolve protein condensates by disrupting hydrophobic interactions[61]. We treated the GFP-BRD4 expressing ARID1A KO HCT116 cells with 1.5% 1,6-HD for 5 min, and observed much reduced BRD4 puncta formation similar to that seen after HERVH knockdown (Fig. 6d). Treatment of the patient-derived CRC organoids with 0.5% 1,6-HD resulted in strong growth inhibition with reduced cell proliferation and increased apoptosis (Supplementary Fig. 6b–d), reminiscing that

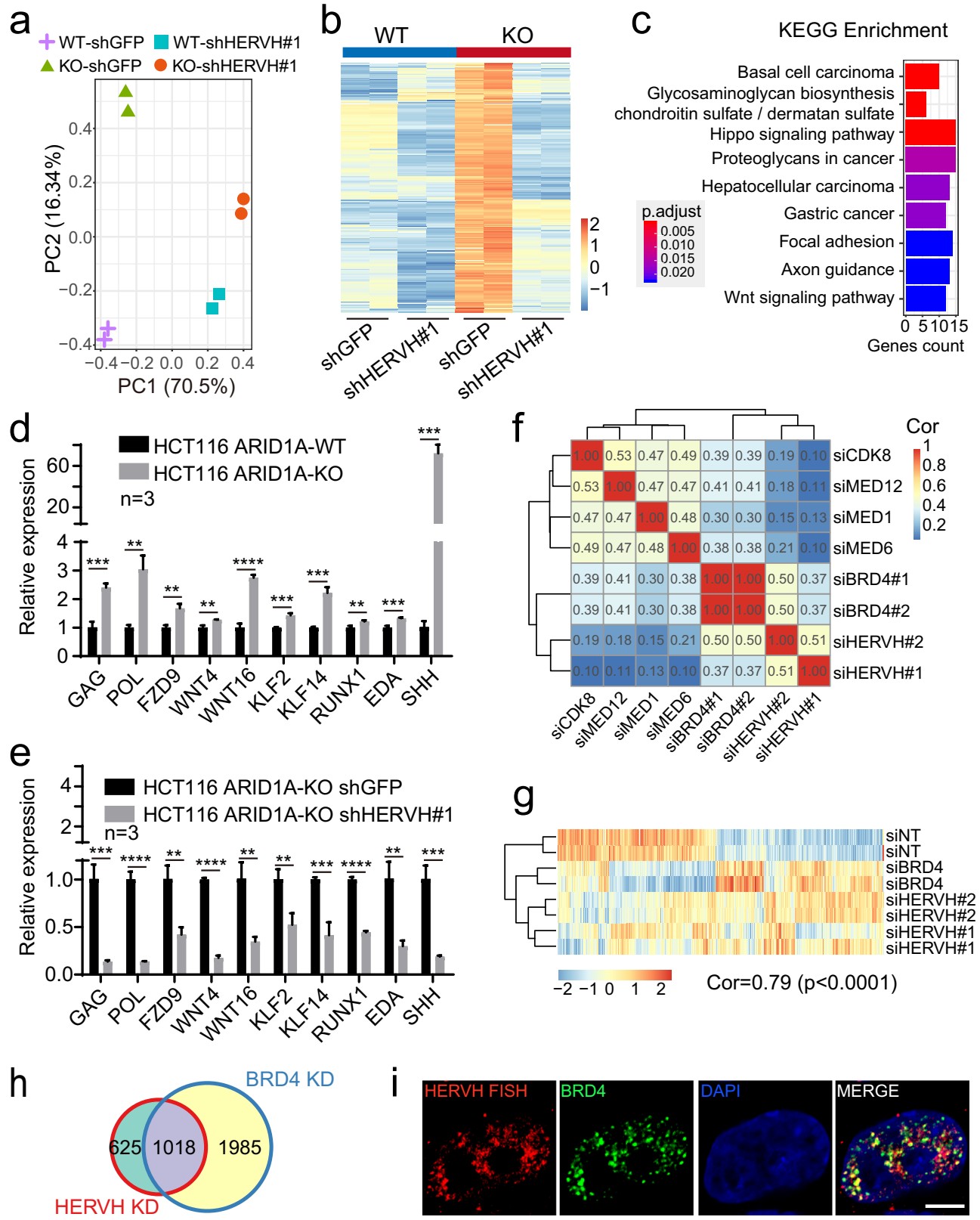

seen with the HERVH shRNA treated organoids. We then compared the influences of HERVH knockdown or 1,6-HD treatment on the genomic distribution of BRD4 using a group of known BRD4 targets identified from a published dataset[62]. Both HERVH knockdown and 1,6-HD treatment attenuated the

binding of BRD4 to the target genes analyzed in the ARID1A KO HCT116 cells (Fig. 6e, f).

It has been reported that ARID1A mutant cells show increased sensitivity to the BET inhibitor JQ1[45,46]. We confirmed that the ARID1A KO HCT116 cells were slightly more sensitive to JQ1 as

**Fig. 5 Functional correlation and colocalization between HERVH RNA and BRD4. a** PCA analysis showing the impacts of HERVH knockdown on the global transcriptome of WT and ARID1A KO HCT116 cells. **b** Heatmap highlighting a set of genes whose expressions increase upon ARID1A loss but decrease again when HERVH is knocked down. **c** KEGG enrichment analysis of the HERVH-dependent genes. Fisher's exact test (one-tailed) is performed. **d, e** qPCR validation of representative HERVH-dependent genes. Data are presented as mean values ± SD from three independent experiments, two-tailed unpaired $t$ test, **$p < 0.01$, ***$p < 0.001$, ****$p < 0.0001$. **f** Correlation matrix showing the unbiased and pairwise comparisons of global transcriptomes upon knockdown of HERVH and the components in mediator complex. Color bar represents Spearman's correlation coefficient (one-tailed test). **g** Heatmap showing strong correlation of changes in transcriptome between knockdowns of HERVH and BRD4 in ARID1A KO HCT116 cells. Spearman's correlation coefficient Cor = 0.79, $p < 0.0001$ (one-tailed test). **h** Venn diagram showing that 1018 genes are coregulated by BRD4 and HERVH. **i** Partial colocalization between HERVH FISH signals and GFP-BRD4 nuclear foci. Results are representative of three independent experiments. Bar: 5 μm. Source data including exact $p$ values are provided as a Source data file.

---

well as another recently reported BRD4 inhibitor NHWD-870[63] (Supplementary Fig. 6e, f). We activated the expression of endogenous HERVH by 2-3 folds using the CRISPR Synergistic Activation Mediator (SAM) approach in WT HCT116 cells (Supplementary Fig. 6g), and we observed a mild increase in sensitivity to the two BET inhibitors (Supplementary Fig. 6h, i), suggesting that the upregulated transcription of HERVH in the ARID1A mutant cells was contributing to the observed increase in sensitivity to BET inhibitors.

## Discussion

Mutational landscape analyses have revealed that ARID1A is among the most frequently mutated epigenetic factors across many cancer types[40,44]. Understanding its mechanism of action and hence identifying targetable vulnerabilities for ARID1A inactivation have been of great importance. In this study, we investigated how the repetitive genome responded to the inactivation of ARID1A and identified a group of *HERVH* elements that were specifically derepressed. This derepression was dependent on ARID1B to some extent and was indispensable for the survival of the CRC cells, likely due to its influence on the dynamics of BRD4 and the regulated transcriptional network (Fig. 6g). Several synthetic lethality targets of ARID1A have been reported, including ARID1B, EZH2, HDAC6, Aurora A, and GCLC[50,54,64–69]. ARID1A mutant cells are also hypersensitive to BET inhibitors[45,46], a promising class of anticancer drugs. Our results imply that the activation of pluripotency-related HERVH is a shared mechanistic foundation of the previously observed ARID1B- and BET vulnerabilities of the ARID1A mutated cells. The HERVH-BRD4 regulatory axis and the adjunct mechanism reported here also offer several potential targets of intervention (Fig. 6g), among which the HERVH itself is of most interest, because of its specific expression in early embryos and general silencing in most adult tissues. It is worth noting that ARID1A has been implicated in several other biological processes, some of which also involve HERVH, such as high-order spatial chromosome partitioning and tissue regeneration[15,16,22,70–72]. The molecular mechanism reported here may have certain explanatory power in those scenarios as well.

Derived from ancient retroviral infections, ERVs are domesticated viral fossils in our genome whose activity is under close surveillance[1,2,4,28]. Comprehensive interrogations in mouse ESCs have revealed that overlapping epigenetic pathways linked to heterochromatin formation are enlisted to suppress the transcription of ERVs. This includes DNA methylation (5-methylcytosine, 5mC), various histone modifications (H3K9me3, H3K27me3, H4K20me3, H4R3me2, and H2AK119ub), and their corresponding writers and readers[31,32]. Reminiscent of the diversity of the process of heterochromatin formation in early embryos[73], different families of ERVs rely on distinct epigenetic means to achieve silencing. The specific recognition of different ERVs by the various epigenetic mechanisms is in part mediated by the KRAB domain-containing zinc finger proteins (KZFPs),

which can bind to specific DNA sequences in individual ERV and recruit KAP1 and other epigenetic modifiers. RNA-mediated targeting mechanisms also contribute to the specific silencing of ERVs. piRNAs as well as other small RNA species are able to bring histone-modifying activities to their complementary ERV loci[4,28,74,75]. Our study reveals another mode of ERVs suppression which involves the SWI/SNF chromatin remodelers, further increasing the complexity of the epigenetic regulatory network constraining the expression of ERVs. The targeting mechanism for the BAF complex in silencing HERVH is currently unknown. It will be interesting to investigate the potential interactions between BAF and KZFPs or the small RNA machineries.

Accumulating evidence reveals that ERVs are co-opted to perform a wide range of biological functions. In early embryos and ESCs, ERVs serve as regulatory elements and alternative promoters to rewire the transcription network of pluripotency[13,21]. Moreover, certain groups of ERVs become transcriptionally activated in an orderly fashion during embryogenesis[33], functioning as enhancer or long noncoding RNAs[14,15], and sometimes synthesizing reverse transcriptase activity and even forming viral-like particles[76]. ERVs are also involved in many human diseases such as various types of cancer. The abnormally activated ERVs can produce long noncoding RNAs or functional polypeptides[37,77–80], enabling cancer cells to exploit and repurpose developmental pathways to promote malignancy[38]. Of particular note, the reactivated ERVs in cancer are extensively recruited as promoters to drive the expression of many oncogenes in a process termed onco-exaptation[81,82]. Our results reciprocally demonstrate that mutations of tumor suppressor can activate functionally important ERVs, suggesting the existence of positive feedback loops between ERVs and cancer driver genes. Future studies shall extend the analysis to other cancer driver genes and characterize these positive feedback loops more comprehensively. The establishment of a mutually reinforcing relationship between cancer driver genes and ERVs will deepen our understanding of the etiology of malignancy and shed light on cancer treatments.

## Methods

**Data download**. The TCGA dataset used in this study, including the RNA-seq BAM files, the gene raw count data (htseq-count files), and the annotated somatic simple nucleotide variation files (MuTect2 VCF) of patients with colon adenocarcinoma (COAD) and rectum adenocarcinoma (READ), were accessed through dbGaP accession number phs000178.v11.p8[48] and downloaded using the gdc-client v1.6.0. The clinical overall survival (OS) information was obtained from Liu et al.[83]. The RNA-seq fastq files GSE50760[84] of normal and tumor tissues from 18 CRC patients were downloaded from https://www.ncbi.nlm.nih.gov/geo/query/acc.cgi?acc=GSE50760. The RNA-seq fastq files of the 59 colorectal cancer cell lines in Cancer Cell Line Encyclopedia[49] (CCLE) were downloaded from https://www.ebi.ac.uk/ena/browser/view/PRJNA523380, and the corresponding germline filtered CCLE merged mutation calls were acquired from https://portals.broadinstitute.org/ccle/data. The previously published RNA-seq and ChIP-seq raw reads fastq files generated with HCT116 cells or mice primary colon epithelial cells[42,50] GSE71514 and GSE101966 were downloaded from https://www.ncbi.nlm.nih.gov/geo/query/acc.cgi?acc=GSE71514 and https://www.ncbi.nlm.nih.gov/geo/query/acc.cgi?acc=GSE101966.

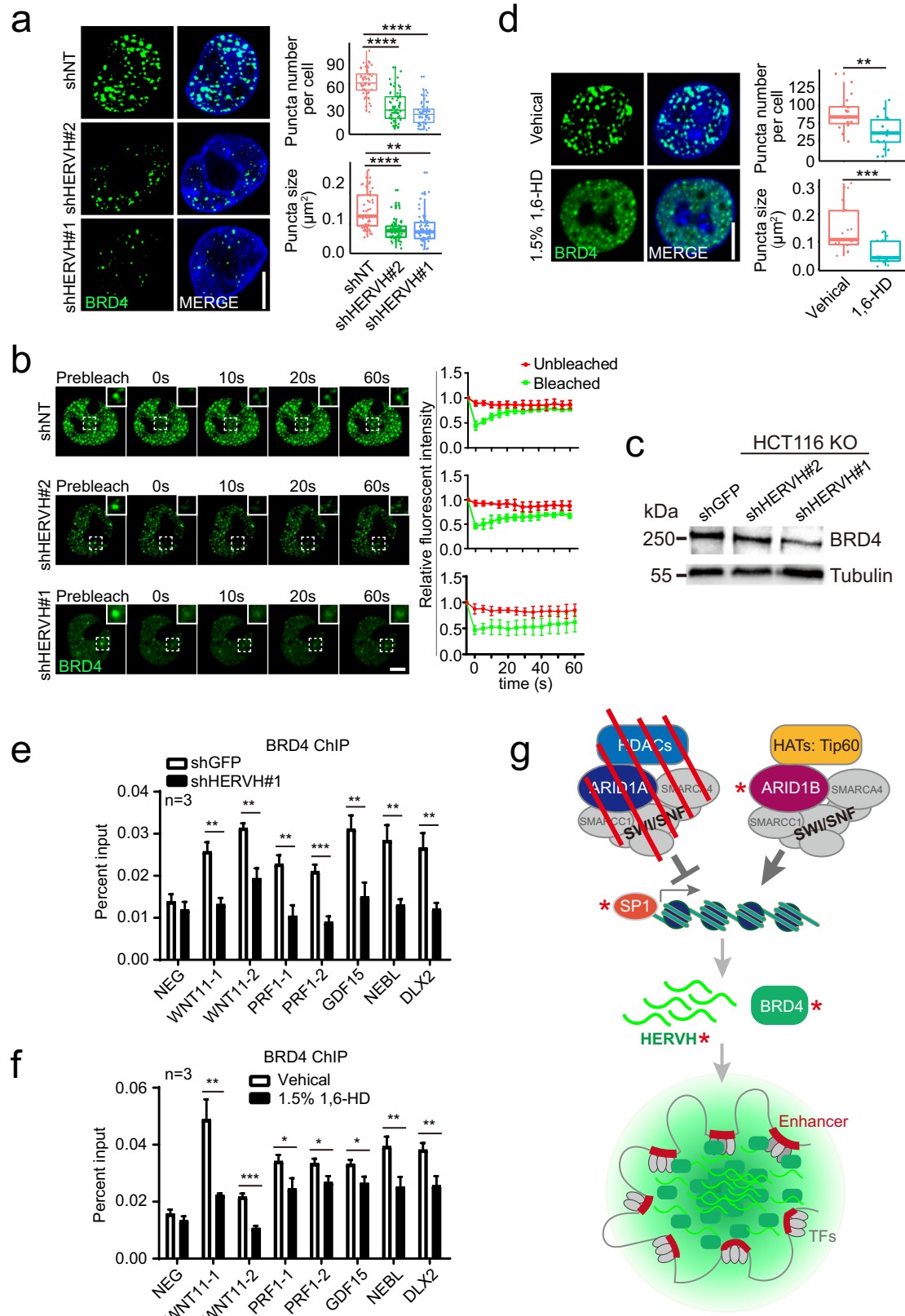

**RNA-seq analysis.** Raw reads were first cleaned using trim_galore v0.6.0 (http://www.bioinformatics.babraham.ac.uk/projects/trim_galore/) with default parameters. The reads from each RNA-seq sample were then mapped to hg38 or mm9 genome assembly downloaded from UCSC, using STAR v2.5.3a[85]. The key alignment parameters were as follows: "--outFilterMismatchNoverLmax 0.04 --outSAMtype BAM SortedByCoordinate --outFilterMultimapNmax 500 --out-MultimapperOrder Random --outSAMmultNmax 1". Gene expression was

quantified using featureCounts v1.6.5[86] of subread-1.6.5 package based on hg38 RefSeq genes annotation file. Repeats expression was quantified using feature-Counts v1.6.5 ("featureCounts --M --fraction") based on repeats annotation files downloaded from https://genome.ucsc.edu/cgi-bin/hgTables. Both multi-mapped reads and uniquely mapped reads were included to calculate the expression at the family level, with the parameter "featureCounts --a annotation.gtf --o sample.-counts --M". To quantify the expression at specific loci, only uniquely mapped

**Fig. 6 HERVH contributes to the formation and dynamics of BRD4 puncta. a** Representative images and quantifications of GFP-BRD4 nuclear foci in control shNT ($n = 52$), shHERVH#2 ($n = 72$), or shHERVH#1 ($n = 71$) treated cells. The box-plot center represents median, the bottom and top lines represent the 25th and 75th percentiles, the whiskers extending to ±1.5× interquartile range (IQR). Data presented are from three independent experiments, two-tailed unpaired $t$ test, $**p < 0.01$, $****p < 0.0001$. Bar: 5 μm. **b** Fluorescence recovery after photobleaching (FRAP) analysis of BRD4 foci after control (NT) or HERVH knockdown. Data are presented as mean values ± SD from at least three independent experiments. Bar: 3.3 μm. **c** Western blot showing decreased BRD4 protein level upon knockdown of HERVH in HCT116 ARID1A KO cells. Results are representative of three independent experiments. **d** Representative images and quantifications of GFP-BRD4 puncta in control vehicle ($n = 27$) or 1.5% 1,6-HD ($n = 26$) treated cells. The box-plot center represents median, the bottom and top lines represent the 25th and 75th percentiles, the whiskers extending to ±1.5× interquartile range (IQR). Data presented are from three independent experiments, two-tailed unpaired $t$ test, $**p < 0.01$, $***p < 0.001$. Bar: 5 μm. **e, f** BRD4 ChIP-qPCR results of a group of known BRD4 target genes in the cells treated with the indicated shRNA or chemical. Data are presented as mean values ± SD from three independent experiments, two-tailed unpaired $t$ test, $*p < 0.05$, $**p < 0.01$, and $***p < 0.001$. **g** A model summarizing how ARID1A loss upregulates HERVH and hence stimulates BRD4 nuclear foci formation and BRD4-mediated transcription. Red asterisks label potential targets of intervention. Source data including exact $p$ values are provided as a Source data file.

reads were included with the parameter "featureCounts --a annotation.gtf --o sample.counts". Principal component analysis was conducted with the functions "vst" and "plotPCA" from R package DESeq2 v1.22.2[87]. Differential expression analysis was performed based on the negative binomial distribution using the functions "DESeq" and "results" from DESeq2. The heatmap of differentially expressed genes or repeats was created using R package pheatmap v1.0.12. The KEGG enrichment analysis was performed using the function "enrichKEGG" from the R package clusterProfiler v3.10.1[88]. Venn diagrams were prepared with the R package Vennerable 3.1.0.9000 and venn 1.10.

**Survival analysis.** The curated clinical endpoint results (OS event and OS event times) of the 628 patients in TCGA-COREAD dataset were obtained from Liu et al.[83]. Only patients in stages II and later according to the American Joint Committee on Cancer (AJCC) pathologic tumor staging system were included. The 493 CRC patients were classified into HERVH-high (145 patients with HERVH-int CPM > 8430.797) and HERVH-low groups (348 patients with CPM < 8430.797), and the survival curves of the two groups were compared using log-rank test from the function "survdiff" in R package survival v2.44-1.1.

**Integration analysis of whole-exome sequencing (WXS) and RNA-seq.** WXS files (MuTect2 VCF) and RNA-seq data from 516 patients in TCGA-COREAD were analyzed (Supplementary Fig. 1a). All the somatic mutational information was included regardless of their classification. For each gene, we classified the patients into WT or mutation group and then calculated the $Log_2$ FC between these two groups using the expression values (CPM) of HERVK-int and HERVH-int. $p$-values were calculated by Wilcoxon test.

**ATAC-seq and ChIP-seq analyses.** Raw reads were cleaned using trim_galore v0.6.0. The reads were then aligned to the hg38 genome assembly using STAR v2.5.3a with the options "--alignIntronMax 1 --alignEndsType EndToEnd" as previously reported[89,90]. The parameter "--outFilterMultimapNmax 1" was applied to include only the uniquely mapped reads. Duplicate reads were then removed using MarkDuplicates from gatk package v.4.1.4.1. Replicate samples were merged using the samtools v1.10[91]. For ATAC-seq, bigwig tracks were generated using bamCoverage from python package deeptools 3.3.1 (parameteres: --skipNAs --normalizeUsing CPM)[92]. For ChIP-seq, bigwig tracks were generate using bamCompare from deeptools 3.3.1 (parameters: --skipNAs --scaleFactorsMethod readCount --operation log2 --extendReads 200). Negative values were set to zero. ATAC-seq and ChIP-seq profiles were created by computeMatrix and plotProfile in deeptools 3.3.1. IGV v.2.4.13 was used to visualize the bigwig tracks[93].

**Cell culture and cell line generation.** The cell lines used in this study, including HCT116, DLD1, SW480, LS174T, SW620, HT29, HCT8, RKO, CRL1790/841, NCM460, FHC, B16F10, HELA, U2OS, and 293T, were cultured in RPMI 1640 or DMEM medium containing 10% FBS and incubated at 37 °C with 5% $CO_2$ in a humidified incubator. E14 mouse embryonic stem cells (mESCs) were cultured on gelatin-coated plates in serum-free ESGRO medium (Millipore, sf001-500p) and passaged every three days in DMEM high glucose (HyClone, SH3002202b) with 15% ESC-qualified fetal bovine serum (Vistech, SE200-ES), 1×non-essential amino acids (NEAA, Gibco, 11140050), 0.1 mM 2-mercaptoethanol (Sigma, m3148), and 1000 U/mL of LIF (Millipore, ESG1107). To generate NCM460 and SW480 ARID1A KO cell lines, the cells were transfected with LentiCRISPR-V2 plasmid carrying sgARID1A (Supplementary Fig. 2b) using Lipofectamine 2000 (Invitrogen) and further selected by 1 μg/mL puromycin (Selleck, s7417) for 3 days. The cells were then plated at single-cell density in 100 mm petri dishes, and the emerged individual clones were picked and replated into 24-well plates. The loss of ARID1A expression was confirmed by western blot. The HCT116 ARID1A KO cell line was generated by knockin of a premature stop codon (Q456*) by gene trap (Horizon Discovery, HD 104-049).

**Organoid culture.** The CRC tissue samples were collected from a 63-year-old man with rectal adenocarcinoma (AJCC stage IIA) at the third Xiangya Hospital according to the principles of Good Clinical Practice and the Declaration of Helsinki, and the CRC organoids were generated as previously described[94]. All the human tissue-related experiments were approved by the Medical Ethics Committee of Central South University, and the informed consent was obtained from the patients. This study is compliant with the Guidance of the Ministry of Science and Technology (MOST) of China for the Review and Approval of Human Genetic Resources (approval no. 2022BC0004). From the resected colon segment, the tumor tissues as well as normal tissues were isolated and stored in ice-cold RPMI 1640 supplemented with 1% Penicillin-Streptomycin. The tissues were then washed in ice-cold DPBS (Biological Industries) supplemented with 1% Penicillin-Streptomycin and cut into 1–3 mm³ cubes. After centrifuging at $200 \times g$ for 5 min, the supernatant was removed and the pellet was resuspended in collagenase IV (Gibco, 17104019) supplemented with 10 μM ROCK inhibitor Y-27632 dihydrochloride (Millipore, SCM075). The tissues were digested at 37 °C for 1 h and mixed up every 10–15 min by pipetting, washed with 10 mL advanced DMEM/F12 (Thermo Fisher Scientific, 12634-010) supplemented with Y-27632, and then centrifuged at $200 \times g$ for 5 min at 4 °C. The pellet was resuspended in DMEM/F12 supplemented with Y-27632 and filtered through 60 μm cell strainer. After centrifugation at $200 \times g$ for 5 min at 4 °C, the supernatant was discarded and the pellet was resuspended in 70% Matrigel (Corning, 356231). 30 μL of the Matrigel mixture was plated on the bottom of 24-well plates, and 500 μL organoid medium (Accurate International Biotechnology, M102-50) was added to each well following incubation at 37 °C with 5% $CO_2$ for 30 min. The organoid medium was changed every 2–3 days, and the organoids were passaged after 7 days of culture.

**Cell growth assays.** For cell viability assays, cells were plated into 96-well plates at the density of 2000–5000 cells per well after infected with lentiviruses expressing shGFP or shHERVHs. The cells were kept for another 7 days, and the viability was measured daily using MTT (Sigma, M5655) as previously described[95]. For chemosensitivity assays, the cells were seeded in 96-well plates and treated with the compounds at indicated concentrations for 72 h, and then the cell viability was measured. For colony formation assays, the cells were seeded at the density of 1000–2000 cells per well in 6-well plates after infected with lentiviruses expressing shGFP or shHERVHs. The cells were allowed to grow for 10–14 days and then fixed for 10 min in 50% (v/v) methanol containing 0.01% (w/v) crystal violet.

**Tumor sphere formation.** The 6-well plates were coated with 12 mg/mL poly (2-hydroxyethyl methacrylate) (poly-HEMA, Sigma, P3932) in 95% ethanol. The indicated cells were digested by TrypLE, and approximately 1000 cells were suspended in 50% Matrigel (Corning, 356231) and plated in the precoated 6-well plates. The 6-well plates containing the cells were incubated at 37 °C for 30 min, and then 2 mL of phenol red-free DMEM/F12 (Gibco, 21041) containing 1 × B27 supplement (Invitrogen, 12587) and 20 ng/mL rEGF (Sigma, E-9644) was added into each well. The culture medium was changed every 2–3 days, and the number of tumor spheres in each well was counted after 12 days.

**Xenograft tumors.** All the mice were housed under the SPF environment with a 12 h light–dark cycle and had a temperature of 22–24 °C with 50–60% humidity. The 4–5 weeks old female BALB/c nude mice were purchased from Hunan SJA Laboratory Animal Co., Ltd. (Changsha, China). HCT116 WT and KO cells were treated with shGFP or shHERVH, and $5\times10^5$ of the indicated cells were suspended in 100 μL DPBS and injected subcutaneously into the flank of the nude mice which were acclimatized for 1 week. Seven days after injection, the tumors were measured twice weekly with an electronic caliper, and the volumes were calculated using the formula: $0.5 \times (length \times width^2)$. The tumor volumes at day 12 were presented (Supplementary Fig. 4g). Measurements were from 6 mice each for the control and HERVH KD WT HCT116 groups, and the results were from 8 and 11 mice, respectively, for the control and HERVH KD ARID1A KO groups. All the animal

experiments were approved by the Medical Ethics Committee of Central South University, and conducted according to the Guidelines of Animal Handling and Care in Medical Research in Hunan Province, China.

**RNA interference**. The siRNA oligos were synthesized by GenePharma (Shanghai GenePharma Co., Ltd.), and the sequences were listed in Supplementary Data 7. Cells were transfected with the indicated siRNA by Lipofectamine 2000 (Invitrogen). After 48 h, the cells were harvested and the efficiency of silencing was verified by qPCR. For shRNA, shRNA oligos were synthesized by Tsingke (Tsingke Biotechnology Co., Ltd.) and cloned into pLKO.1 TRC Cloning vector (Supplementary Data 7). The shRNA and packaging vectors (pMD2.G and psPAX2) were transiently co-transfected into 293T cells by polyethylenimine (Sigma, P3143), and the resulted lentivirus particles were harvested and precipitated by PEG8000. The target cells were treated with lentivirus particles and 8 μg/mL polybrene for 24 h, and the efficacy of shRNA interference was determined by qPCR. The knockdown efficiency and statistics for each experiment were listed in Supplementary Data 8.

**Treatment of organoids with shRNA or chemicals**. For shRNA-mediated knockdown, the organoids cultured in Matrigel were washed once with DPBS, and digested with TrypLE for 5 min at 37 °C. During the digestion, Matrigel was disrupted by pipetting repeatedly. When cell clumps containing 2–10 cells were observed, 10 mL of advanced DMEM/F12 was added before centrifugation at $200 \times g$ for 5 min. The supernatant was removed and the cells were resuspended using organoid medium supplemented with 8 μg/mL polybrene. Then the cells were split equally into 2 wells of 24-well plate precoated with poly-HEMA, and 50 μL of lentivirus carrying shGFP or shHERVH was added. After spin infection at 2000 rpm for 1 h, the cells were incubated at 37 °C with 5% CO$_2$ for 4 h. The cells were then resuspended in 10 mL of advanced DMEM/F12 and centrifuged at $200 \times g$ for 5 min. The pellet was resuspended with 100 μL of 70% Matrigel, and 10 μL of the mixture was plated per well into prewarmed 96-well plate. The organoids were cultured for 10–14 days and the medium was changed every 2–3 days. For 1,6-hexanediol treatment, the organoids were digested and plated in 96-well plate. After growing for 3 days, the organoids were treated with vehicle or 0.5% 1,6-hexanediol for 7 days. The organoid medium containing vehicle or 0.5% 1,6-hexanediol was changed every 2 days.

**Endogenous HERVH activation by CRISPR SAM**. Endogenous HERVH was activated using the CRISPR-Cas9 Synergistic Activation Mediator (SAM) system. HCT116 cells were infected with the lentiviruses of lenti dCAS-VP64_Blast (Addgene #61425) and lenti MS2-p65-HSF1_Hygro (Addgene #61426). Forty-eight hours later, the cells were treated with blasticidin and hygromycin for 7 days. Lentivirus containing sgRNA-NT or sgRNA-LTR7Y was used to infect these cells followed by puromycin selection for 5 days, and then the activation efficiency was detected by qPCR.

**Western blot**. Cells were washed with cold DPBS for two times and then lysed in 2× Laemmli buffer (2% SDS, 20% glycerol, and 125 mM Tris-HCl, pH 6.8) supplemented with 1× protease inhibitor cocktail (Sigma, P8340). The cell lysate was scraped and sonicated, and the concentration of protein was determined by BCA assay. The protein was separated by SDS-PAGE and transferred onto nitrocellulose membrane. The membrane was then blocked with 5% non-fat milk for 1 h at room temperature, and incubated with the indicated primary antibody overnight at 4 °C with shaking. The membrane was washed for 3 times and incubated with secondary antibodies for 2 h. The signal was then detected with ECL substrates (Millipore, WBKLS0500). Dilutions of antibodies were: rabbit anti-ARID1A Ab (1:1000, Abcam, ab182560), rabbit anti-BRD4 Ab (1:1000, Active Motif, 39909), mouse anti-α-Tubulin Ab (1:3000, Cell Signaling, 3873 s). Goat anti-Rabbit IgG (H + L) Secondary Antibody (1:5000, Thermo Fisher Scientific, 31460), Goat anti-Mouse IgG (H + L) Secondary Antibody (1:5000, Thermo Fisher Scientific, 31430). The full scan blots were included in the Source data file.

**RNA-seq and qPCR**. The RNA of the treated cells was extracted by TRIzol (Life Technologies, 87804) according to the manufacturer's protocol. Total RNA was made into libraries for sequencing using the mRNA-Seq Sample Preparation Kit (Illumina) and sequenced on an Illumina Hiseq platform (Novagene, Tianjin, China). The sequencing data were deposited into the GEO database (accession number GSE180475). For RT-qPCR, RNA was extracted by TRIzol, and reverse transcribed to cDNA using the PrimeScript RT reagent Kit (Takara, RR037A). The cDNA was then used as templates and qPCR was performed using the SYBR Green qPCR Master Mix (SolomonBio, QST-100) on the QuantStudio 3 Real-Time PCR system (Applied Biosystems). Primers used in qPCR were listed in Supplementary Data 7.

**Chromatin immunoprecipitation**. HCT116 ARID1A WT and KO cells in 100 mm petri dishes were cross-linked with 1% formaldehyde for 10 min at room temperature and quenched with 125 mM ice-cold glycine. For the ChIP experiments after 1,6-hexanediol treatment, the cells were treated with 1.5% 1,6-hexanediol or control vehicle for 30 min before cross-linking. The indicated cells were then rinsed

with 5 mL ice-cold 1 × PBS for two times, and harvested by scraping using silicon scraper. After spinning at 1350 g for 5 min at 4 °C, the supernatant was discarded, and the pellet was resuspended in Lysis Buffer I (50 mM HEPES-KOH, pH 7.5, 140 mM NaCl, 1 mM EDTA, 10% glycerol, 0.5% NP-40, 0.25% Triton X-100 and 1 × protease inhibitors) and incubated at 4 °C for 10 min with rotating. After spinning at 1350 g for 5 min at 4 °C, the pellet was resuspended in Lysis Buffer II (10 mM Tris-HCl, pH 8.0, 200 mM NaCl, 1 mM EDTA, 0.5 mM EGTA, and 1× protease inhibitors), incubated for 10 min at room temperature, and spun at $1350 \times g$ for 5 min at 4 °C. The pellet was again resuspended in Lysis Buffer III (10 mM Tris-HCl pH 8.0, 100 mM NaCl, 1 mM EDTA, 0.5 mM EGTA, 0.1% Na-Deoxycholate, 0.5% N-lauroylsarcosine and 1× protease inhibitors) and transferred into Covaris microTUBEs. The DNA was sonicated to 200 bp fragments using Covaris S220 (duty cycle: 10; intensity: 4; cycles/burst: 200; duration: 200 s). After quenching the SDS by 1% of Triton X-100, the lysate was spun at $20,000 \times g$ for 10 min at 4 °C. 50 μL of supernatant from each sample was reserved as input, and the rest lysate was incubated overnight at 4 °C with the magnetic beads bound with ARID1A (Cell Signaling, 12354 S), ARID1B (Santa Cruz, sc-32762X), SMARCA4 (Abcam, ab110641) or H3K27ac (Abcam, ab4729) antibody respectively. The beads were washed three times with Wash Buffer (50 mM Hepes-KOH, pH 7.6, 500 mM LiCl, 1 mM EDTA, 1% NP-40, 0.7% Na-deoxycholate), and washed once with 1 mL TE buffer containing 50 mM NaCl. The DNA was eluted with 210 μL of Elution Buffer (50 mM Tris-HCl, pH 8.0, 10 mM EDTA, 1% SDS). The cross-links were reversed by incubated at 65 °C overnight. 200 μL of TE buffer was added to each tube, and the RNA was degraded by incubation with 16 μL of 25 mg/mL RNase A at 37 °C for 60 min. The protein was degraded by adding 4 μL of 20 mg/mL proteinase K and incubating at 55 °C for 60 min. The DNA was then purified by phenol:chloroform:isoamyl alcohol extraction, and resuspended in 50 μL ddH$_2$O. The fragments of HERVH DNA were detected by qPCR using primers specifically targeting the HERVH_3 locus (Supplementary Data 7). The ChIP-qPCR primers were designed based on the ChIP-seq dataset generated in this study (GSE180475).

**The RNAscope™ in situ hybridization (ISH)**. The colon cancer tissue array (HCol-Ade180Sur) was purchased from Shanghai Biochip Co. Ltd (Shanghai, China). The RNAscope analysis with probes targeting the HERVH-gag sequence was performed using the RNAscope Multiplex Fluorescent Reagent Kit v2 (ACD bio, 323100) according to the manufacturer's protocol. The HERVH consensus sequence used for probe design was listed in Supplementary Data 10. Following the RNAscope staining, the tissue array was imaged with a LSM880 confocal microscope (Zeiss).

**RNA-FISH combined with immunofluorescence**. RNA-FISH combined with immunofluorescence was performed as previously described[59]. Cells cultured on poly-L-lysine-coated coverslips were fixed with 10% formaldehyde in DPBS for 10 min. After three washes in DPBS, cells were permeabilized with 0.5% Triton-X100 for 10 min. The cells were then washed three times in DPBS and blocked with 4% Bovine Serum Albumin for 30 min. The cells were incubated with the indicated primary antibody diluted in DPBS overnight, washed three times in DPBS, and incubated again with the secondary antibody for 1 h. After washing twice with DPBS, the cells were fixed again with 10% formaldehyde in DPBS for 10 min. Following two washes with DPBS, the cells were further washed in Wash Buffer I (20% Stellaris RNA FISH Wash Buffer A (Biosearch Technologies, Inc., SMF-WA1-60), 10% Deionized Formamide (Invitrogen, AM9342) in RNase-free water) for 5 min. The RNA probe (Stellaris) in hybridization buffer was added to the cells and incubated at 37 °C for 16 h. After washing with Wash Buffer I at 37 °C for 30 min, the cells were stained with 1 μg/mL DAPI for 5 min. The cells were then washed with Wash Buffer B (Biosearch Technologies, Inc., SMF-WA1-60) for 5 min, and rinsed once in water before mounting with SlowFade Diamond Antifade Mountant (Invitrogen, S36963). Dilutions of primary antibodies were: rabbit anti-BRD4 (1:500, Active Motif, 39909), rabbit anti-MED1 (1:500, Abcam, ab64965), rabbit anti-MED12 (1:500, Bethyl, A300-774A), rabbit anti-CDK8 (1:500, Active Motif, 61481), Goat anti-Rabbit IgG (H + L) Cross-Adsorbed Secondary Antibody, Alexa Fluor™488 (1:500, Thermo Fisher Scientific, A11008). The sequence of the RNA probe (Stellaris) was listed in Supplementary Data 6.

**Immunofluorescence with organoids**. The immunofluorescence of organoids was performed as previously described[96]. The organoids cultured in 96-well plate were washed once with DPBS without disrupting the Matrigel, and then 200 μL of ice-cold cell recovery solution (Corning, 354253) was added and incubated at 4 °C for 1 h with shaking at 60 rpm. After the Matrigel was dissolved, the organoids were rinsed out using ice-cold PBS with 1% BSA and spun down at $70 \times g$ for 3 min at 4 °C. The pellet of organoids was resuspended in 1 mL of 10% formaldehyde in DPBS, and incubated at 4 °C for 45 min. 9 mL of ice-cold PBT (0.1% Tween 20 in DPBS) was added and incubated at 4 °C for 10 min. The organoids were then spun down at 70 g for 5 min at 4 °C, resuspended in 200 μL ice-cold OWB (0.1% Triton X-100, 0.2% BSA in DPBS), and transferred into 24-well plate precoated with poly-HEMA. Following incubation at 4 °C for 15 min, 200 μL of the indicated primary antibody diluted in OWB was added and incubated overnight at 4 °C with shaking at 60 rpm. The next day, 1 mL of OWB was added into each well. After all the organoids were settled at the bottom of the well, the OWB was removed with just

200 µL left in each well. The organoids were washed three times with 1 mL of OWB and incubated at 4 °C for 2 h with shaking at 60 rpm. The OWB was removed with just 200 µL left in each well, and then 200 µL of secondary antibody diluted at 1:200 in OWB was added and incubated overnight at 4 °C with shaking at 60 rpm. After the incubation, the organoids were washed once with OWB, and 200 µL of 2 µg/mL DAPI in OWB was added and incubated at 4 °C for 30 min. The organoids were then washed two times with OWB, transferred to 1.5-mL Eppendorf tube, and spun down at $70 \times g$ for 3 min at 4 °C. The OWB was removed as much as possible without touching the organoids, and the organoids were resuspended with fructose-glycerol clearing solution (60% glycerol and 2.5 M fructose in ddH$_2$O). Drew a $1 \times 2$ cm rectangle in the middle of a slide, and placed 3 layers of sticky tape at both sides of the rectangle. The organoids were transferred into the middle of the rectangle, and put the coverslip on the top. The images were taken with a LSM880 confocal microscope (Zeiss). Dilutions of primary antibodies were: rabbit anti-E-Cadherin (1:400, Cell Signaling, 3195 S), mouse anti-Ki67 (1:400, Cell Signaling, 9449 S), Goat anti-Rabbit IgG (H + L) Cross-Adsorbed Secondary Antibody, Alexa Fluor™647 (1:400, Thermo Fisher Scientific, A21244), Goat anti-Mouse IgG (H + L) Cross-Adsorbed Secondary Antibody, Alexa Fluor™568 (1:400, Thermo Fisher Scientific, A11004).

**Fluorescence recovery after photobleaching**. The treated cells were plated into 35 mm glass-bottom confocal dishes (NEST, 801001), and the FRAP experiment was performed on the Zeiss LSM880 Airyscan confocal microscope with a ×63 Plan-Apochromat 1.4 NA oil objective. The Zeiss TempModule system was used to control the temperature (37 °C), the humidity, and the CO$_2$ (5%) of the imaging system. After imaging for 3 frames, the cells were photo-bleached using 100% laser power with the 488 nm laser (iterations: 50, stop when intensity drops to 50%). The cells were then imaged again every 2 s. The images were analyzed and measured with ZEN 2.3 blue edition (Zeiss) and Image J v1.48.

**Statistical analysis**. The experiments were conducted in at least three independent biological replicates, and the data were presented as mean ± SD. If not specified, the Student's $t$ test was used to perform a statistical significance test between different groups, and $p < 0.05$ was considered significant. All statistical and correlation analyses were performed using the GraphPad Prism 8.0 software (GraphPad Software, Inc.).

**Reporting summary**. Further information on research design is available in the Nature Research Reporting Summary linked to this article.

## Data availability
The data that support this study are available from the corresponding author upon reasonable request. All the sequencing data have been deposited to the GEO database with the accession number GSE180475. The previously published RNA-seq and ChIP-seq fastq files GSE50760, GSE71514, and GSE101966 are also available on GEO. CCLE RNA-seq fastq files PRJNA523380 are available on ENA. Source data are provided with this paper.

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

## Acknowledgements

We gratefully acknowledge Professor François Mallet for HERVH antibodies and Professor Xiang Chen and Professor Mingzhu Yin for BET inhibitors. We thank Dr. Kai Fu and Dr. Joong Sup Shim for cell lines and colleagues in the center of medical genetics at Central South University and members of the Yuan lab for helpful discussions. This project has been supported by the National Natural Science Foundation of China (grants 31771589, 91853108, 92153301, and 32170821 to K.Y., 81801426 to L.S., and 32101034 to F.C.), Department of Science & Technology of Hunan Province (grants 2017RS3013, 2017XK2011, 2018DK2015, 2019SK1012, and 2021JJ10054 to K.Y, 2019JJ40478 to P.L., 2021JJ41049 to C.Y, and the innovative team program 2019RS1010), and Central South University (2018CX032 to K.Y., 2019zzts339 to X.L., and the innovation-driven team project 2020CX016). K.Y. is supported by the National Thousand Talents Program for Young Outstanding Scientists.

## Author contributions

C.Y., X.L., X.H., P.L., and K.Y. conceived the project. X.H. initiated this project. C.Y., X.L., F.C., S.M., L.L., H.L., X.H., R.W., L.S., N.Z., Y.M., Y.S., and P.L. performed the experiments. X.L., J.C., and S.M. performed all the bioinformatic and statistical analyses with the supervision of K.Y. C.L. assisted with patient selection. S.H. and Z.Z. provided insightful suggestions in project design and manuscript writing. K.Y. wrote the manuscript with help from C.Y. and X.L.

## Competing interests

The authors declare no competing interests.
