## [Peer Review File · Nature Communications]

ARID1A loss derepresses a group of human endogenous retrovirus-H loci to modulate BRD4-dependent transcriptionREVIEWER COMMENTS

Reviewer #1 (Remarks to the Author):

In this study, it is reported that, in colorectal cancer, loss of ARID1A, which is a known tumor suppressor, is associated with increased transcription of some copies of the highly repetitive HERVH family of endogenous retroviruses. This family appears to be most sensitive, compared to the 100's of other ERV families. This upregulation is dependent upon ARID1B. A range of experiments are performed to show that HERVH transcripts appear to colocalize with nuclear BRD4 loci and are involved in regulating some genes. Inhibition of HERVH by shRNA appears to greatly slow the growth of CRC cells in culture and as organoids, suggesting a potential therapeutic target. This is an interesting and, if verified, an important study and the authors were quite comprehensive in their analyses. However, there are some concerns that need to be addressed:

1. In the Introduction, it would be prudent to mention that approximately half of the "2000 copies" of HERVH are in fact simply solitary LTRs.
2. In figure S4D, the dramatic effect on viability/growth of multiple CRC cell lines upon treatment with shRNA for HERVH is very striking but needs to be substantiated to ensure it is not non-specific toxicity. (This also pertains to similar experiments in figure 4 on growth of organoids, etc.) The same experiments in Fig. S4D should be done with a few cell lines that do not express HERVH. One would predict there would be no effect on such cell lines. Also, the effects of treating with the other shRNA (HERVH #2) should be shown. Another way of demonstrating that the viability/growth effect is truly due to HERVH would be to overexpress a HERVH construct (in the presence of the shRNA) with a few point mutations to render it immune from shRNA KD and show that the growth is rescued. Since a major conclusion of the authors is that inhibition of HERVH expression impairs tumor growth, it is critical that non-specific toxicity be eliminated from the possibilities.
3. In figure 3, panels E, F and G, please explain the x-axes. Does the "zero" point in the middle represent the entire HERVH element or just the 5' end of it? This should be made clear. For example, in figure 3E, does the ATAC-seq peak in the KO cells downstream of "00" occur within the interior of the HERVH element or downstream of it?
4. In Figures 3F and G, why are the ChIP-seq profiles for H3K27ac and H3K4me in the KO cells uniformly higher across the 100 kb span centering on HERVH? This broad "distance effect" is not commented upon in the text. As far as I am aware, these histone marks do not typically spread to such distances. How were the results normalized? Given the uniformly higher level in the KO cells, the authors need to perform pairwise comparisons genome-wide for histone marks, perhaps in 10kb bins, to see if the data, processed as they do, yields genome-wide increases in the two marks in the KO. If so, this is a red-flag for the way the data was analyzed and might throw the interpretation into question.
5. Also, why is there a peak of H3K27ac in Figure 3F approximately 25 kb upstream of the HERVH element in both the WT and KO cells?
6. The supplementary tables downloaded from the journal web site were not labeled with their Table number, making it very difficult to examine them.
7. It is unclear how the list of potential TFs to test was derived. Are these just the TFs that have predicted binding sites in HERVH? In just the LTR or anywhere in the element? It is interesting that Sp1 was identified as being important for HERVH expression. The authors

should reference this paper that first reported an effect of Sp1 on HERVH expression 25 years ago (PMID: 8659116).

8. From figure S3D, it seems only a minor fraction of the ~1000 HERVH elements (excluding solitary LTRs) are derepressed in HCT116 cells when ARID1A is knocked out. Specifically, only 20 elements are in common to both datasets. What is different about these 20 compared to the rest? Did the authors examine sequence differences or genomic context? Is it fair to say that HERVH is derepressed when only 20 of 1000 elements are affected?

Reviewer #2 (Remarks to the Author):

This manuscript explores the underlying properties within LTR/ERVs in the human genome, with a focus on colorectal cancers and normal tissue. The authors find a number of ERVs that are either down or up regulated in CRC relative to normal controls. A number of these were validated using RNA-Scope probes and by focusing on a specific element called HERVH, expression was correlated with mutational status of a handful of genes, including ARID1A, the BAF subunit. By taking isogenically matched cell line models, the authors show that ARID1A knockout resulted in changes in specific ERVs and this was validated in additional models of CRC. Specific mouse ERVs were shown to be differential expressed when ARID1A was deleted genetically. ARID1B was shown to play a regulatory role, specifically under conditions where ARID1A was depleted, suggesting a redundancy role. Various epigenetic marks and factors were characterized at the HERVH locus and an analysis of potential transcription factors, revealed Sp1 as the best candidate. The phenotypic role of HERVH is explored and it is shown to be essential for viability of CRC in multiple different assays. The ARID1A dependent genes were shown to be partially dependent on HERVH and the physical location of HERVH with BRD4 and Mediator subunits were explored and some cursory experiments are conducted to link HERVH with BRD4 activity. Finally, the recent observation that ARID1A depleted lines show dependence to BRD inhibitors was validated in a CRC cell line.

This paper explores a potentially interesting topic and there is a lot of data in the paper, but much of it is superficial and key functional conclusions are not explored in sufficient detail.

- Sp1 is implicated, but does Sp1 bind to the HERVH regulatory locus? Just showing that Sp1 can regulate HERVH doesn't show that it is the direct transcription factor. Does modulation of Sp1 change BRD4 function or anything else at the HERVH locus?
- ARID1B is implicated in the absence of ARID1A, but no global ARID1B studies are conducted. Does ARID1B bind to the HERVH locus in the absence of ARID1A? At the moment, published genomic data is used to show changes (albeit modest in some cases) of various marks and proteins at the HERVH locus in the absence of ARID1A (Figure 3E-3H). However, there is no evidence that these have anything to do with ARID1B. Do the changes in H3K27Ac or ATAC-seq occur in the absence of ARID1B?
- The authors suggest that this regulation and role of HERVH is cancer specific. Does the depletion of HERVH do anything to a normal, non-cancerous colorectal model?
- The authors do not explore the link between HERVH and BRD4 in sufficient detail. Is BRD4 binding affected by the presence or absence of HERVH? At the moment the authors claim "The results reported here suggested that the upregulated transcription of HERVH in ARID1A mutant cells contributed to the formation of BRD4 nuclear puncta and stimulated their dynamic activity (Fig. 5K), providing an explanation for the observed increased sensitivity to BET inhibitors". However, there is no evidence showing a functional link? Does modulation of HERVH change the responsiveness to BRD inhibitors? If not, then it is

correlative speculation that there is a functional connection.

- Figure 4D. The authors don't state the number of mice used, but since this is the only in vivo experiment, the information needs to be clear.
- The paper has lots of typos and small errors.

Reviewer #3 (Remarks to the Author):

Yu et al report that knock out (KO) of ARID1A in CRC cells leads to transcriptional activation of HERVH, which is required for the survival of CRC cells. Furthermore, they demonstrated that the HERVH transcripts contribute to BRD4 protein nuclear puncta formation and are important for BRD4's function, providing a mechanism that how HERVH contributes to CRC cell proliferation.

In sum, this study is interesting, while lacking some necessary clarification and quality control in experiments, attenuating its conclusions. For example, they did not show the strategy to knockout ARID1A and the confirmation of knockout like WB/RNA-seq. Moreover, such as the absence of the quantification of HERVH Knock-down efficiency, and no rescue experiment for shARID1B, also hinder proper interpretation of the data. Besides, it falls short in providing solid links between the main observations. For example, it is unclear how ARID1A suppresses HERVH, and whether the puncta-formation of BRD4 is important for its function on CRC cell proliferation.

Here are some concerns to highlight:

1. What are the multiple bands in ARID1A immunoblots for Figure 2I? The authors need to add a schematic plot for ARID1A KO strategy to explain why ARID1A expression is still detected in the KO cell line (Figure 2J).
2. For the NCM460 samples in Figure 2G-H, the upregulation magnitude of HERVH is quite incremental while the P-values are very significant, suggesting the replicates are nearly identical. The legend indicates the data getting from two different primer sets. The authors should clarify whether the data is from technical replicates or from independent biological replicates to avoid any misinterpretation.
3. For repetitive genome ChIP-seq/ATAC-seq analysis, the sequencing data has been processed to include multi-mapping reads. Whilst this is an acceptable approach to draw conclusions at the family level, it is not appropriate when looking at individual copies/loci (given that multi-mapping reads are randomly allocated). Multi-mapping reads can be included to draw a family-wide trend as in the line plots (Figure 3E-G), but they should not be depicted as individual loci (Figure 3H). It is important for the authors to display analysis for uniquely mapped reads for individual copies, and add the line plot with all mapped reads (unique + multiple) for family level, and be completely clear in the text/figure legends whenever multi-mapped reads are included.
4. Figure S3 A-C, as ChIP-qPCR for the repetitive sequence is often biased with the primer, the ChIP-seq should be performed. This is important as the authors claim ARID1A/ARID1B/SMARCA4 binds to HERVH directly.
5. Figure 4, the qPCR results of KD efficiency of HERVH should be shown.
6. In Figure S4F, it seems that many cells are not expressed HERVH, and most HERVH transcripts are not located in the nuclear. Conversely, In Figure 5H, most HERVH transcripts are located in the nuclear, why they are inconsistent? Besides, the author should add the statistic results like Figure 5I to avoid bias.
7. The HERVH transcript contribute to the formation of BRD4 puncta is quite interesting, alternatively, this part somewhat disjointed with other observations of this study, as it is still unclear how BRD4-HERVH puncta promote tumorigenesis and still unknown if this process is BRD4 phase separation dependent.

REVIEWER COMMENTS

Reviewer #1 (Remarks to the Author):

In this study, it is reported that, in colorectal cancer, loss of ARID1A, which is a known tumor suppressor, is associated with increased transcription of some copies of the highly repetitive HERVH family of endogenous retroviruses. This family appears to be most sensitive, compared to the 100's of other ERV families. This upregulation is dependent upon ARID1B. A range of experiments are performed to show that HERVH transcripts appear to colocalize with nuclear BRD4 loci and are involved in regulating some genes. Inhibition of HERVH by shRNA appears to greatly slow the growth of CRC cells in culture and as organoids, suggesting a potential therapeutic target. This is an interesting and, if verified, an important study and the authors were quite comprehensive in their analyses.

We want to thank the reviewer for recognizing the importance of our study. According to the reviewers' comments, we have performed additional experiments and analyses to refine and consolidate our main conclusions, meanwhile, we have thoroughly edited the manuscript to clarify the ambiguities and make it more readable for a broad readership.

However, there are some concerns that need to be addressed:

1. In the Introduction, it would be prudent to mention that approximately half of the "2000 copies" of HERVH are in fact simply solitary LTRs.

We have modified this description accordingly. The sentence now reads: "Like other human ERVs, most of the HERVH elements are no longer intact but truncated forms and solitary LTRs, and only approximately 100 copies are close to full-length."

2. In figure S4D, the dramatic effect on viability/growth of multiple CRC cell lines upon treatment with shRNA for HERVH is very striking but needs to be substantiated to ensure it is not non-specific toxicity. (This also pertains to similar experiments in figure 4 on growth of organoids, etc.) The same experiments in Fig. S4D should be done with a few cell lines that do not express HERVH. One would predict there would be no effect on such cell lines. Also, the effects of treating with the other shRNA (HERVH #2) should be shown. Another way of demonstrating that the viability/growth effect is truly due to HERVH would be to overexpress a HERVH construct (in the presence of the shRNA) with a few point mutations to render it immune from shRNA KD and show that the growth is rescued. Since a major conclusion of the authors is that inhibition of HERVH expression impairs tumor growth, it is critical that non-specific toxicity be eliminated from the possibilities.

We want to thank the reviewer for raising this important issue. As the reviewer has pointed out, a main conclusion of our study is that inhibition of HERVH suppresses the growth of CRC cells. It is indeed critical to rule out the possible off-target toxicity, and we have eased this concern from two different angles. Firstly, we have repeated the key experiments with two additional shRNAs targeting different regions of HERVH (Fig. 4A). All the three shRNAs treatments resulted in similar growth impairment of CRC cells and organoids. We have updated Fig. 4 and Fig. S4 accordingly to include some of the new results. Secondly, we have expanded the

HERVH knockdown experiment to additional cell lines, including the mice mESCs and B16F10 which do not have HERVH elements in its genome, as well as human cancer cell lines from other tissues (HELA and U2OS). Indeed, the mice cells showed no growth inhibition when treated with HERVH shRNAs. Interestingly, the cervical cancer cell line HELA and the osteosarcoma cell line U2OS, despite expressing HERVH to varying degrees, also showed minimum sensitivity toward HERVH knockdown (Fig. S4). The underlying mechanism for this intriguing tissue specificity is currently unknown, and we have launched another project to address it. Lastly, we wanted to perform the HERVH rescue experiment as the reviewer suggested, however, as a metagene, the HERVH family comprises a group of evolutionarily related elements harboring different mutations and truncations. Therefore, it is not as feasible as coding genes (like we have added for the ARID1B knockdown experiment) in terms of designing a fully functional rescue construct.

3. In figure 3, panels E, F and G, please explain the x-axes. Does the “zero” point in the middle represent the entire HERVH element or just the 5’ end of it? This should be made clear. For example, in figure 3E, does the ATAC-seq peak in the KO cells downstream of “00” occur within the interior of the HERVH element or downstream of it?

We have updated and rearranged the panels and now they are in Fig S3A-C. The two zero points “0 0” in the previous figure were indeed the start and the end of the HERVH elements. We have changed the labeling and included more panels to show signals both around and within the HERVH elements.

4. In Figures 3F and G, why are the ChIP-seq profiles for H3K27ac and H3K4me in the KO cells uniformly higher across the 100 kb span centering on HERVH? This broad “distance effect” is not commented upon in the text. As far as I am aware, these histone marks do not typically spread to such distances. How were the results normalized? Given the uniformly higher level in the KO cells, the authors need to perform pairwise comparisons genome-wide for histone marks, perhaps in 10kb bins, to see if the data, processed as they do, yields genome-wide increases in the two marks in the KO. If so, this is a red-flag for the way the data was analyzed and might throw the interpretation into question.

We have reanalyzed the ChIP-seq data and updated these panels (according to the suggestions of Reviewer 3), and now they are in Fig. S3B-C. The ChIP-seq signals were normalized with default parameters of the bamCompare tool from the python package deeptools (see Method, ATAC-seq and ChIP-seq analyses section). In the updated Figure S3, we have included the results from the genome-wide comparisons in 10kb bins as suggested by the reviewer, and we have also presented the signals at the rest of the HERVH loci that were not activated in the KO cells. Both results showed comparable baseline levels of the ChIP-seq signals. Although we still do not know why the regions around the activated HERVH elements had higher signals of these two histone marks, it was not an artifact caused by our data analysis process.

5. Also, why is there a peak of H3K27ac in Figure 3F approximately 25 kb upstream of the HERVH element in both the WT and KO cells?

We want to thank the reviewer for bringing up this interesting point. We have

examined all the activated HERVH loci individually, and identified one major contributor on chr6, where we found many peaks upstream of the activated HERVH locus. These peaks seemed to be on both the regulatory sequences and the gene body of the UTRN gene (see the attached figure 1 below), which was also upregulated in the ARID1A KO cells. HERVH elements can have both trans and cis effects on gene expressions. In this study we mainly focused on their influence on the global transcriptome via HERVH RNA, however, they could also function by regulating adjacent gene transcription. Therefore, we have initiated a separate study to look into this possible regulatory role of HERVH on the UTRN gene.

Figure 1. Genomic snapshot showing the HERVH locus and the upstream UTRN gene. Note that the expression level of HERVH_9 was significantly higher than that of UTRN (CPM ~16000 versus ~30), making the UTRN mRNA signals hard to see in the snapshot.

6. The supplementary tables downloaded from the journal web site were not labeled with their Table number, making it very difficult to examine them.

We have updated the supplementary tables accordingly.

7. It is unclear how the list of potential TFs to test was derived. Are these just the TFs that have predicted binding sites in HERVH? In just the LTR or anywhere in the element? It is interesting that Sp1 was identified as being important for HERVH expression. The authors should reference this paper that first reported an effect of Sp1 on HERVH expression 25 years ago (PMID: 8659116).

The list of potential TFs in the previous manuscript was derived from Ito J et al. 2017 PLoS Genet. (PMID: PMC5529029), in which they collected and analyzed the publicly available ChIP-seq datasets of 97 TFs, and identified the TFs in the list that were able to bind the HERVH-int and LTR elements. We analyzed the expression of these TFs in the HCT116 cells, and selected the ones with high transcription levels.

We want to thank the reviewer for reminding us this important paper on SP1. And, we have changed the description accordingly to emphasize this previous discovery. Now it reads “Gain of SP1 sites is associated with transcriptional activation of HERVH loci⁵⁶. To verify if SP1 was involved in the activation of the subset of HERVH elements in ARID1A mutated HCT116 cells, we knocked down its expression by siRNA and performed RNA-seq analysis. Two additional transcription factors (TF) that were predicted to bind HERVH were included for comparison¹¹. Among the three TFs analyzed, only SP1 knockdown significantly reduced the expression of HERVH.”

8. From figure S3D, it seems only a minor fraction of the ~1000 HERVH elements (excluding solitary LTRs) are derepressed in HCT116 cells when ARID1A is knocked out. Specifically, only 20 elements are in common to both datasets. What is different about these 20 compared to the rest? Did the authors examine sequence differences or genomic context? Is it fair to say that HERVH is derepressed when only 20 of 1000 elements are affected?

The existence of multiple copies of HERVH elements in the human genome indeed complicates the related analyses. Both this reviewer and Reviewer #3 raised this important issue on the specific derepression of only a fraction of HERVH elements. According to the suggestions, we have reanalyzed the sequencing data using two different strategies. When characterizing the expression at the family level, both multi-mapped reads and uniquely mapped reads were included; when quantifying the expression or histone modifications at specific loci, only the uniquely mapped reads were included. The details of the procedures were added in the Method. This refinement in analysis strategies slightly altered the specific results but did not change the overall conclusions. With the new strategies, we identified the HERVH loci that were unambiguously derepressed in ARID1A KO cells (Fig. 3F), and we depicted the chromatin accessibility as well as the histone modifications on these loci (Fig. S3A-C). Additionally, inspired by the previous point raised by the reviewer, we have compared the density of SP1 binding motifs between the derepressed HERVH group and that remained silenced, and interestingly, we found that the former harbored more SP1 binding motifs. This might contribute to the locus-specific derepression observed here, and we have added this result to the revised manuscript (Fig. S3H). We agree with the reviewer that it is not so fair to conclude that HERVH as a whole is derepressed. Therefore, we have changed the related texts, particularly, we have changed the title to “ARID1A loss derepresses a group of human endogenous retrovirus-H loci to modulate BRD4-dependent transcription”. We think these changes have significantly improved the manuscript and we are grateful to the reviewers for these suggestions.

Reviewer #2 (Remarks to the Author):

This manuscript explores the underlying properties within LTR/ERVs in the human genome, with a focus on colorectal cancers and normal tissue. The authors find a number of ERVs that are either down or up regulated in CRC relative to normal controls. A number of these were validated using RNA-Scope probes and by focusing on a specific element called HERVH, expression was correlated with mutational status of a handful of genes, including ARID1A, the BAF subunit. By taking isogenically matched cell line models, the authors show that ARID1A knockout

resulted in changes in specific ERVs and this was validated in additional models of CRC. Specific mouse ERVs were shown to be differentially expressed when ARID1A was deleted genetically. ARID1B was shown to play a regulatory role, specifically under conditions where ARID1A was depleted, suggesting a redundancy role. Various epigenetic marks and factors were characterized at the HERVH locus and an analysis of potential transcription factors, revealed Sp1 as the best candidate. The phenotypic role of HERVH is explored and it is shown to be essential for viability of CRC in multiple different assays. The ARID1A dependent genes were shown to be partially dependent on HERVH and the physical location of HERVH with BRD4 and Mediator subunits were explored and some cursory experiments are conducted to link HERVH with BRD4 activity. Finally, the recent observation that ARID1A depleted lines show dependence to BRD inhibitors was validated in a CRC cell line.

We want to thank the reviewer for this thorough summary of our main findings.

This paper explores a potentially interesting topic and there is a lot of data in the paper, but much of it is superficial and key functional conclusions are not explored in sufficient detail.

We are pleased that the reviewer found the topic of our study interesting. According to the suggestions, we have done additional experiments and analyses to better support our main conclusions. We have also streamlined the manuscript to make it more focused.

- Sp1 is implicated, but does Sp1 bind to the HERVH regulatory locus? Just showing that Sp1 can regulate HERVH doesn't show that it is the direct transcription factor. Does modulation of Sp1 change BRD4 function or anything else at the HERVH locus?

It has been reported that gain of SP1 sites is associated with transcriptional activation of HERVH loci, and SP1 can bind the LTR of HERVH in gel mobility shift assays (Nelson D.T. et al. 1996 Virology. PMID: 8659116). ChIP-seq results also show SP1 bindings to the HERVH elements (Ito J. et al. 2017 PLoS Genet. PMID: PMC5529029), suggesting that it is involved in HERVH transcription as a TF. We have analyzed the density of SP1 motifs and observed that the derepressed HERVH loci harbored more SP1 motifs than the HERVH elements that remained silenced (Fig. S3H). We have also checked the BRD4 dynamics after SP1 knockdown according to the reviewer's suggestion, and in addition to the markedly reduced HERVH expression (Fig. S3F-G), we observed similar reduction of BRD4 puncta to that seen in the HERVH knockdown cells (Fig. S6A). These results support its function as the main TF in HERVH transcription in the HCT116 KO cells.

- ARID1B is implicated in the absence of ARID1A, but no global ARID1B studies are conducted. Does ARID1B bind to the HERVH locus in the absence of ARID1A? At the moment, published genomic data is used to show changes (albeit modest in some cases) of various marks and proteins at the HERVH locus in the absence of ARID1A (Figure 3E-3H). However, there is no evidence that these have anything to do with ARID1B. Do the changes in H3K27Ac or ATAC-seq occur in the absence of ARID1B?

It has been reported that ARID1B, by supplying residual BAF complex activities, is essential for the survival of ARID1A mutated cancer cells (Kelso T.W.R. et al. 2017 *Elife*. PMID: PMC5643100, Helming K.C. et al. 2014 *Nat Med*. PMID: PMC3954704, Trizzino M. et al. 2018 *Cell Rep*. PMID: PMC6146183). This has drawn our attention to ARID1B in regulating HERVH in the cells lacking ARID1A. Both the RNA-seq data and our qPCR results with two different shRNAs targeting ARID1B showed that the expression level of HERVH was indeed sensitive to ARID1B knockdown (Fig. 3A-D). Moreover, in the additional rescue experiment, we showed that re-expression of a mutant ARID1B resistant to shRNA treatment restored the HERVH expression levels (Fig. 3E). These results unambiguously support the conclusion that ARID1B contributes to the transcription of HERVH in the absence of ARID1A. We have rearranged the panels in Fig. 3 and S3, and thoroughly edited the related texts to emphasize this point.

Given that both ARID1A and ARID1B function as the rigid structure core in the BAF complex in a mutually exclusive way, we speculated an increased binding of ARID1B in the absence of ARID1A. To test this, we have attempted the global ChIP-seq analyses of both ARID1A and ARID1B using commercially available antibodies (ARID1A: CST #12354; ARID1B: Santa Cruz, sc-32762X). However, we found the quality of the ChIP-seq data not satisfying, with relatively high background and compromised specificity genome-wide. We did however observe a mild increase of ARID1B on the derepressed HERVH loci compared to the HERVH elements that remained silenced (see the attached figure 2 below). Alternatively, because the HERVH_3 locus was most highly activated in the ARID1A KO cells, we selected it to characterize the local changes by ChIP-qPCR using two different primer sets (Fig. 3G). We observed that the amount of ARID1A on HERVH_3 was indeed reduced in ARID1A KO cells, and the binding of ARID1B to this region was increased compensatorily (Fig. 3H-J).

Figure 2. ARID1B ChIP-seq results showing a slight increase of occupancy on the derepressed HERVH loci.

As suggested by the reviewer, we analyzed the changes in chromatin accessibility and the histone modifications after ARID1B knockdown. We observed that although ARID1B KD reduced the HERVH expression, it did not influence the epigenetic signatures analyzed here (Fig. S3A-C). However, we could not draw the conclusion that it was independent of ARID1B, because ARID1B is essential for the viability of the ARID1A mutated cells (Kelso T.W.R. et al. 2017 *Elife*. PMID: PMC5643100, Helming K.C. et al. 2014 *Nat Med*. PMID: PMC3954704, Trizzino M. et al. 2018 *Cell Rep*. PMID: PMC6146183), which greatly limited the KD efficiency (65% KD efficiency in the analyzed dataset). Meanwhile, we did not know whether the changes in the histone modifications were causal or consequential. So, we have moved these results to Fig. S3 and changed the description to make it more accurate.

- The authors suggest that this regulation and role of HERVH is cancer specific. Does the depletion of HERVH do anything to a normal, non-cancerous colorectal model?

We want to thank the reviewer for bringing up this important issue. As a main conclusion of our study is about the specific role of HERVH transcripts for CRC cells, it is crucial to eliminate the non-specific toxicity, which was also pointed out by Reviewer #1. To this end, we have repeated the related experiments with two additional shRNAs targeting different regions of HERVH (Fig. 4A), and we found that all the three shRNAs treatments resulted in similar phenotypes (Fig. 4). Moreover, we have performed the HERVH knockdown experiments with additional cell lines, including the mice mESCs and B16F10 which do not have HERVH elements in its genome. These cells as expected showed no growth inhibition when treated with HERVH shRNA. We have also included non-cancerous cells of colorectal origin such as FHC in our assay, and we still observed a response to HERVH knockdown, although the sensitivity was reduced compared with other CRC cells (Fig. S4A). Surprisingly, when we treated the cervical cancer cell line HELA and the osteosarcoma cell line U2OS with HERVH shRNA, we observed very limited effects (Fig. S4C). This difference was not entirely due to the expression levels of HERVH, but rather reflected certain tissue specificity, the mechanism of which is currently unknown and we are actively pursuing it in a separate project.

- The authors do not explore the link between HERVH and BRD4 in sufficient detail. Is BRD4 binding affected by the presence or absence of HERVH? At the moment the authors claim “The results reported here suggested that the upregulated transcription of HERVH in ARID1A mutant cells contributed to the formation of BRD4 nuclear puncta and stimulated their dynamic activity (Fig. 5K), providing an explanation for the observed increased sensitivity to BET inhibitors”. However, there is no evidence showing a functional link? Does modulation of HERVH change the responsiveness to BRD inhibitors? If not, then it is correlative speculation that there is a functional connection.

We want to thank the reviewer for pointing out this critical issue. We have performed a series of experiments to further explore the functional link between HERVH and BRD4. Firstly, we investigated the influence of HERVH knockdown on the genomic distribution of BRD4 by ChIP-qPCR using a group of known BRD4 targets identified from a published dataset (Baranello L. et al. 2016 Cell. PMID: PMC4826470), and the results showed that the binding of BRD4 to its targets was compromised after HERVH knockdown (Fig. 6E). Secondly, to test if the impact of HERVH knockdown on BRD4 binding was due to the reduced puncta formation, we used 1,6-hexanediol to chemically disrupt the BRD4 condensates, and we observed quite similar results in terms of BRD4 binding to its target genes (Fig. 6F). Lastly, to explore if modulation of HERVH changes the responsiveness to BETi, we activated the expression of endogenous HERVH by 2-3 folds using the CRISPR Synergistic Activation Mediator (SAM) system in WT HCT116 cells, and we indeed observed a slight increase in sensitivity (Fig. S6G-I). However, we do not think there is a simple linear relationship between the HERVH expression level and the responsiveness to BETi.

- Figure 4D. The authors don't state the number of mice used, but since this is the only in vivo experiment, the information needs to be clear.

Unlike genes, ERVs are quite diverse between primates and rodents, which limits the use of mice model in the related studies. Therefore, we rely largely on the patient-derived organoids. We have added the number of mice used in the figure legends as well as in the Method section, and we have included more experimental details for the mice study.

- The paper has lots of typos and small errors.

We have thoroughly edited the manuscript as suggested.

Reviewer #3 (Remarks to the Author):

Yu et al report that knock out (KO) of ARID1A in CRC cells leads to transcriptional activation of HERVH, which is required for the survival of CRC cells. Furthermore, they demonstrated that the HERVH transcripts contribute to BRD4 protein nuclear puncta formation and are important for BRD4's function, providing a mechanism that how HERVH contributes to CRC cell proliferation. In sum, this study is interesting, while lacking some necessary clarification and quality control in experiments, attenuating its conclusions.

We appreciate that the reviewer recognizes the significance of our study. We have performed additional control experiments and analyses to refine and consolidate our main conclusions, and we have thoroughly edited the manuscript to clarify the ambiguities.

For example, they did not show the strategy to knockout ARID1A and the confirmation of knockout like WB/RNA-seq.

We have added the schematics and the corresponding descriptions for the generation of ARID1A KO cell lines (Fig. S2A-B). We in fact confirmed the KO cells using western blot in the previous submission, which is shown in Fig. 2F in the revised manuscript.

Moreover, such as the absence of the quantification of HERVH Knock-down efficiency, and no rescue experiment for shARID1B, also hinder proper interpretation of the data.

We have added a separate supplementary table 8 to list all the knockdown efficiencies and statistics. We have also performed the rescue experiment for ARID1B KD as suggested by the reviewer (Fig. 3E).

Besides, it falls short in providing solid links between the main observations. For example, it is unclear how ARID1A suppresses HERVH, and whether the puncta-formation of BRD4 is important for its function on CRC cell proliferation.

We want to thank the reviewer for raising these two issues. To directly address how ARID1A suppresses HERVH was quite challenging, we instead investigated the involvement of ARID1B in activating the HERVH. Both the RNA-seq data and our qPCR results with two different shRNAs targeting ARID1B showed that the expression level of HERVH was downregulated after ARID1B knockdown (Fig. 3A-D). Moreover, we showed that re-expression of a mutant ARID1B resistant to

shRNA treatment restored the HERVH expression level (Fig. 3E). We further showed that the amount of ARID1B on the activated HERVH locus was compensatorily increased in the ARID1A KO cells using ChIP-qPCR (Fig. 3H-J). So, we concluded that ARID1B contributes to the transcriptional activation of a subset of HERVH loci in the absence of ARID1A. And now the question becomes why ARID1A and ARID1B show opposite activities. This phenomenon in fact has been documented and analyzed in previous studies (Wang X. et al. 2004 *Biochem J.* PMID: PMC1134073, Nagl N.G. et al. 2007 *EMBO J.* PMID: PMC1794396). Both ARID1A and ARID1B function as a rigid core in the BAF complex, but they are mutually exclusive and show opposite roles in cell cycle control. Molecular characterization has revealed that ARID1A- and ARID1B-containing BAF complexes are associated with quite different histone acetyltransferase (HAT) and deacetylase (HDAC) activities, providing a compelling explanation for the observed opposite functions. In the previous manuscript, we have tested chemical inhibitors targeting the HAT and HDACs, and we indeed observed the expected changes in HERVH expression. However, we realized that the effects of these inhibitors could be very broad, so that we have removed the results from the revised manuscript, and we have changed the descriptions of these results to put the emphasis on the involvement of ARID1B, a conclusion that is better supported by the data.

We have also explored whether the puncta formation of BRD4 is functionally important as suggested by the reviewer. We used 1,6-hexanediol to chemically disrupt the BRD4 condensates and compared the consequences with that of HERVH knockdown. Using ChIP-qPCR on a group of known BRD4 target genes, we found that both HERVH knockdown and 1,6-hexanediol treatment attenuated the binding of BRD4 to its targets (Fig. 6E-F). We also treated the CRC organoids with low dose of 1,6-hexanediol, and we observed strong growth inhibition with reduced cell proliferation and increased apoptosis (Fig. S6B-S6D), reminiscing that seen with the HERVH shRNA treated organoids. These new results to some extent suggest that the puncta formation of BRD4 is of functional importance.

Here are some concerns to highlight:

1. What are the multiple bands in ARID1A immunoblots for Figure 2I? The authors need to add a schematic plot for ARID1A KO strategy to explain why ARID1A expression is still detected in the KO cell line (Figure 2J).

The ARID1A antibody we used in Western blot is from Abcam (ab182560). The multiple bands detected in the immunoblots seemed to be the degradation products of ARID1A, as documented in several previous reports (Helming K.C. et al. 2014 *Nat Med.* PMID: PMC3954704, Watanabe R. et al. 2014 *Cancer Res.* PMID: 24788099, Dong X. et al. 2021 *Gut.* PMID: 33785559, Shen J. et al. 2015 *Cancer Discov.* PMID: PMC4497871). We have added the schematics and the corresponding descriptions for the generation of ARID1A KO cell lines (Fig. S2A-B).

2. For the NCM460 samples in Figure 2G-H, the upregulation magnitude of HERVH is quite incremental while the P-values are very significant, suggesting the replicates are nearly identical. The legend indicates the data getting from two different primer sets. The authors should clarify whether the data is from technical replicates or from independent biological replicates to avoid any misinterpretation.

The data was from independent biological replicates. We have repeated the experiments and confirmed the observation. For clarity, we have added a separate supplementary table 8 to list all the knockdown efficiencies and statistics, and a statistical analysis paragraph has also been added in the Method section.

3. For repetitive genome ChIP-seq/ATAC-seq analysis, the sequencing data has been processed to include multi-mapping reads. Whilst this is an acceptable approach to draw conclusions at the family level, it is not appropriate when looking at individual copies/loci (given that multi-mapping reads are randomly allocated). Multi-mapping reads can be included to draw a family-wide trend as in the line plots (Figure 3E-G), but they should not be depicted as individual loci (Figure 3H). It is important for the authors to display analysis for uniquely mapped reads for individual copies, and add the line plot with all mapped reads (unique + multiple) for family level, and be completely clear in the text/figure legends whenever multi-mapped reads are included.

We are grateful to the reviewer for this key technical suggestion. We have reanalyzed all the genomic data according to this comment, and updated most of the panels in Fig. 3 and S3. Specifically, we used both multiple mapped reads and uniquely mapped reads to characterize repeats expression at the family level without referring to individual genomic locus. We generated panels in Fig. 1, Fig. S1, Fig. 2, Fig. S2, Fig. 3A-3C, Fig. S3G following this protocol. Meanwhile, we used only the uniquely mapped reads to quantify the expression or other epigenetic marks at specific genomic loci, and the panels Fig. 3F, 3G, S3A-S3C, S3F were generated accordingly. The details of the procedures were added in the Method. This refinement in analysis strategies slightly altered the specific results but did not change the overall conclusions. With the new strategies, we identified the HERVH loci that were unambiguously derepressed in ARID1A KO cells (Fig. 3F), and we depicted the chromatin accessibility as well as the histone modifications on these loci (Fig. S3A-C). These changes, as suggested by the reviewer, have no doubt improved the accuracy of the analyses and the manuscript.

4. Figure S3 A-C, as ChIP-qPCR for the repetitive sequence is often biased with the primer, the ChIP-seq should be performed. This is important as the authors claim ARID1A/ARID1B/SMARCA4 binds to HERVH directly.

As suggested by the reviewer, we have performed the ChIP-seq experiments of both ARID1A and ARID1B using commercially available antibodies (ARID1A: CST #12354; ARID1B: Santa Cruz, sc-32762X). However, the overall quality of the ChIP-seq data of these ARID proteins was poor and the background signals were pretty high, perhaps due to the dynamic nature of the chromatin remodelers or possible cross reactions with other proteins of the ARID family. In fact, many previous studies didn't perform genome-wide ChIP-seq experiments as well, one of them specifically mentioned the lack of suitable antibodies in such experiment (Bitler B.G. et al. 2015 Nat Med. PMID: PMC4352133, Mathur R. et al. 2017 Nat Genet. PMID: PMC5285448, Kelso T.W.R. et al. 2017 Elife. PMID: PMC5643100, Helming K.C. et al. 2014 Nat Med. PMID: PMC3954704, Fukumoto T. et al. 2018 Cell Rep. PMID: PMC5903572).

We have therefore tackled this problem in an alternative way. By quantifying the

uniquely mapped reads as described above, we have identified a group of HERVH elements that were specifically activated in ARID1A KO cells. Given a full length HERVH comprises HERVH-int and the flanking LTRs, we have merged these interspersed HERVH elements according to their genomic locations into 13 bigger clusters (Fig. 3F), among which the HERVH_3 was most highly activated in ARID1A KO cells and close to a full-length HERVH. We have decided to select the HERVH_3 for more detailed characterizations of the local epigenetic changes. We designed two sets of primers to probe the occupancies of ARID1A, ARID1B, SMARCA4, as well as the H3K27ac histone modification (Fig. 3H-3J, S3D-S3E). Although the background signals still existed in the ARID1A ChIP-qPCR experiments (the residual signals in the ARID1A KO cells), we were able to detect that at this specific locus the amount of ARID1A was indeed reduced in ARID1A KO cells, and the binding of ARID1B to this region was increased compensatorily (Fig. 3H-J).

5. Figure 4, the qPCR results of KD efficiency of HERVH should be shown.

We have added a separate supplementary table 8 to list all the knockdown efficiencies and statistics.

6. In Figure S4F, it seems that many cells are not expressed HERVH, and most HERVH transcripts are not located in the nuclear. Conversely, In Figure 5H, most HERVH transcripts are located in the nuclear, why they are inconsistent? Besides, the author should add the statistic results like Figure 5I to avoid bias.

The results shown in Fig. S4F in the previous submission were from the patient-derived organoids, which we found more difficult to work with in the FISH experiment. The results in Fig. 5F were from cultured cells. Given that the purpose of Fig. S4F was to validate the knockdown efficiency of HERVH in organoids, we have removed the results from the revised manuscript, instead, we have presented the qPCR results to better display the knockdown efficiencies (Fig. S4I).

7. The HERVH transcript contribute to the formation of BRD4 puncta is quite interesting, alternatively, this part somewhat disjointed with other observations of this study, as it is still unclear how BRD4- HERVH puncta promote tumorigenesis and still unknown if this process is BRD4 phase separation dependent.

We have edited and streamlined the manuscript to make it more coherent. By comparing the impacts of HERVH knockdown and BRD4 knockdown on the global transcriptome, we have uncovered the significant correlation between HERVH and BRD4 (Fig. 5F-5H, S5B), suggesting that they function in concert to promote tumorigenesis. However, to directly ascribe this function to the BRD4-HERVH puncta was quite challenging, and we have taken an indirect way to explore this. 1,6-hexanediol is widely used to dissolve protein condensates by disrupting hydrophobic interactions. We treated the GFP-BRD4 expressing ARID1A KO HCT116 cells with 1.5% 1,6-hexanediol for 5 min, and observed much reduced BRD4 puncta formation similar to that seen after HERVH knockdown (Fig. 6D). Therefore, we compared the biological consequences of the 1,6-hexanediol treatment and HERVH KD. Using ChIP-qPCR on a group of known BRD4 target genes, we found that both HERVH knockdown and 1,6-hexanediol treatment attenuated the binding of BRD4 to its targets (Fig. 6E-F). We also treated the CRC organoids with low dose of

1,6-hexanediol, and we observed strong growth inhibition with reduced cell proliferation and increased apoptosis (Fig. S6B-S6D), reminiscing that seen with the HERVH shRNA treated organoids. Although still correlative, these new results suggest that the BRD4-HERVH puncta contributes to tumorigenesis.

REVIEWERS' COMMENTS

Reviewer #1 (Remarks to the Author):

The authors have adequately addressed my major concerns and the manuscript is improved.

Reviewer #2 (Remarks to the Author):

The authors have addressed all of my concerns. The new data experimentally addresses the various issues I raised.

Reviewer #3 (Remarks to the Author):

I appreciate the authors' efforts in responding to the previous reviews, and the revisions have improved the manuscript.

For the previous question 4, the authors did not reply to my question, published ARID1A and ARID1B ChIP-seq data are available (PMID: 28925401, PMID: 26716708, ENCODE Project Consortium), and also plenty of published SMARCA4 ChIP-seq data. For Figure 3G-J, the qPCR primers are inside the HERVH element, which can't represent the HERVH_3 element in Figure 3G. And I did not find the qPCR primer sequence in the supplementary table, the authors need to make sure the qPCR primer only target to this locus.

And I do concern about the data quality. In Figure 5 and Figure 6, the authors only show the result of shHERVH#1, while according to Figure 4G, shERVH#1/#2/#3 showed different performance. At least two independent cell lines need to be included to avoid any biased observations.

For Figure 5H &G, I am confused why need to change to siRNA, and the siHERVH#1/#2 showed low correlation.

Overall, the phenomenon reported by the authors are interesting, but more strict quality control will expand my enthusiasm for this paper.

Manuscript Number: NCOMMS-21-26577A

REVIEWER COMMENTS

Reviewer #1 (Remarks to the Author):

The authors have adequately addressed my major concerns and the manuscript is improved.

Reviewer #2 (Remarks to the Author):

The authors have addressed all of my concerns. The new data experimentally addresses the various issues I raised.

Reviewer #3 (Remarks to the Author):

I appreciate the authors' efforts in responding to the previous reviews, and the revisions have improved the manuscript.

We are pleased that all the three reviewers find our manuscript improved, and we appreciate the efforts that the reviewers have put into the assessment.

For the previous question 4, the authors did not reply to my question, published ARID1A and ARID1B ChIP-seq data are available (PMID: 28925401, PMID: 26716708, ENCODE Project Consortium), and also plenty of published SMARCA4 ChIP-seq data.

We might not be explicit enough in the previous response, but we indeed performed the suggested ChIP-seq experiments and replied to the question. Taking into consideration the new comment from the reviewer, we have done a careful literature search and collected published ChIP-seq data for ARID1A/ARID1B/SMARCA4, including the ones mentioned by the reviewer (Table 1). The ChIP-seq data from the papers listed by the reviewer (PMID: 28925401, PMID: 26716708, and PMID: 22955616 in the ENCODE Project Consortium, highlighted with green background in Table 1) were not generated with colorectal cancer cell lines. In fact, we failed to identify any ARID1A or ARID1B ChIP-seq data that was generated with HCT116 cells, except that Mathur R et al. performed SMARCA4 ChIP-seq in their *Nat Genet* paper (PMID: 27941798). The data was already used in our manuscript (GSE71514, Supplementary Fig. 3f).

Table 1 List of collected ARID1A/ARID1B/SMARCA4 ChIP-seq data.

Cell line	Origin	ARID1A ChIP-seq	ARID1B ChIP-seq	SMARCA4 ChIP-seq	PubMed UniqueIdentifier
HCT116	Colorectal cancer	Yes	Yes	Yes	This study
HepG2	Liver cancer	Yes	Yes	No	PMID: 26716708
LNCaP	Prostate cancer	Yes	No	Yes	PMID: 28925401
K562	Leukemia	No	Yes	Yes	PMID: 22955616
HCT116	Colorectal cancer	No	No	Yes	PMID: 27941798
MCF7	Breast cancer	Yes	No	Yes	PMID: 31913353
RMG-1	Ovarian cancer	Yes	No	No	PMID: 29949775
RMG-1	Ovarian cancer	Yes	No	No	PMID: 31131328
12Z	Endometriotic epithelial cells	Yes	No	No	PMID: 31391455
A549	Lung cancer	Yes	No	No	PMID:33627422

During the previous revision, we analyzed our own ChIP-seq data and tried to plot the distributions of ARID1A and ARID1B on HERVH loci. As shown in the following Figure 1a, ARID1A was indeed moderately enriched on the derepressed HERVH loci compared to ARID1B in WT HCT116 cells (upper panels, red lines). And, in ARID1A KO HCT116 cells, the ARID1A signal on the derepressed HERVH loci was reduced, accompanying with a noticeable increase of ARID1B ChIP-seq signal (lower panels, red lines). The trends were consistent with our conclusion. However, we observed a significant amount of signal with the ARID1A antibody in the ARID1A KO cells, suggesting a high background in the ChIP-seq experiment. This was perhaps due to the dynamic nature of the chromatin remodelers or possible cross reactions with other proteins of the ARID family. For the sake of clarity, we have therefore left out the genome-wide ChIP-seq results in the revised manuscript, but only included the results of ChIP-qPCR specifically targeting the most highly activated *HERVH_3* locus. The ChIP-seq data was used to guide the design of the ChIP-qPCR primers (mentioned in the Methods section). We have uploaded these ChIP-seq data of ARID1A and ARID1B along with our RNA-seq data to the GEO database (GSE180475) for readers who might be interested.

Figure 1 ChIP-seq analyses of the distributions of the BAF complex components on the HERVH loci. The indicated ChIP-seq signals at the derepressed HERVH loci were plotted in red. For comparison, signals from the whole genome (10kb bins) or the rest HERVH loci that remain repressed were plotted in blue or green respectively.

Taking one step further, we have also analyzed several ARID1A ChIP-seq data from other cancer cell lines (Table 1, highlighted in red) to see if ARID1A can bind to the derepressed *HERVH* loci (Figure 1b). Interestingly, the enrichment of ARID1A on these genomic loci varied markedly among different cancer cell lines, reminiscing the tissue specificity seen in the cell viability experiments upon *HERVH* knockdowns (Supplementary Fig. 4). The mechanism is unclear and we think it is worth future investigation.

For Figure 3G-J, the qPCR primers are inside the *HERVH* element, which can't represent the *HERVH_3* element in Figure 3G. And I did not find the qPCR primer sequence in the supplementary table, the authors need to make sure the qPCR primer only target to this locus.

We had in fact included the qPCR primers in the previous supplementary Table 7 (lines 75-76, HERVH3-1 and HERVH3-2). Despite the overall sequence similarities, there are enough nucleotide differences among HERVH loci that can be used to design specific primers for a given element. We double checked the specificity of the qPCR primers by NCBI's Primer-BLAST, and each primer set identified only one target site on chromosome 1 in the human genome (GRCh38.p13, Chr1: 68385954-68386054 and Chr1: 68386010-68386111 respectively). Additionally, the melting curve from the qPCR showed only one peak, also validating the specificity of the primers.

And I do concern about the data quality. In Figure 5 and Figure 6, the authors only show the result of shHERVH#1, while according to Figure 4G, shERVH#1/#2/#3 showed different performance. At least two independent cell lines need to be included to avoid any biased observations.

The reviewer raised concerns about the data quality, because HERVH shRNAs showed different performance in some experiments, particularly when applied to the CRC organoids (Fig. 4g). This is in fact an interesting issue specific to repetitive elements, and we should have elaborated this in the previous response. More than 20 different *HERVH* loci were derepressed in the ARID1A KO HCT116 cells and contributing to the production of HERVH transcripts. Due to the nucleotide differences among individual HERVH loci, different HERVH shRNA targeted a distinct subset of the derepressed HERVH elements (Table 2), therefore generating similar but not identical phenotypes. We have added an explanation into the Results section of our manuscript, which says “of note, given that the HERVH transcripts were produced from multiple evolutionarily related genomic loci, different shRNAs demonstrated certain selectivity toward different subsets of the HERVH elements”. We have also included the differential targeting information in supplementary Data 7.

Table 2 The selectively targeted HERVH loci by different shRNAs.

Derepressed HERVH loci	shHERVH#1 (siRNA#1)	shHERVH#2	shHERVH#3	shHERVH#4 (siRNA#2)
HERVH_3 (int_dup1)	yes	yes	yes	yes
HERVH_9 (int_dup1)	yes	yes		
HERVH_9 (LTR_dup2)	yes	yes		
HERVH_3 (int_dup4)	yes	yes	yes	yes
HERVH_3 (int_dup3)	yes	yes	yes	yes
HERVH_3 (int_dup5)	yes	yes	yes	yes
HERVH_3 (LTR_dup2)	yes	yes	yes	yes
HERVH_3 (int_dup2)	yes	yes	yes	yes
HERVH_3 (int_dup6)	yes	yes	yes	yes
HERVH_9 (int_dup6)	yes	yes	yes	yes
HERVH_9 (LTR_dup1)	yes	yes		
HERVH_9 (int_dup3)	yes	yes		
HERVH_3 (LTR_dup1)	yes	yes		
HERVH_9 (int_dup4)	yes	yes		
HERVH_9 (int_dup2)	yes	yes		
HERVH_9 (int_dup5)	yes	yes		

HERVH_2 (int)				
HERVH_12 (int)	yes	yes	yes	
HERVH_10 (LTR)				
HERVH_1 (int)				
HERVH_13 (int)		yes		
HERVH_11 (int)	yes	yes		
HERVH_4 (int)	yes	yes		yes
HERVH_7 (int)				
HERVH_6 (LTR)				
HERVH_5 (int)				
HERVH_8 (int)		yes	yes	
Number of HERVH target loci	19/27	21/27	10/27	9/27
Target location	HERVH-pro	HERVH-pro	HERVH-pro	HERVH-gag

In terms of the different performances of shRNAs on organoids, the varying knockdown efficiency could be another contributing factor. Indeed, using qPCR primers targeting HERVH-gag, the knockdown efficiency for shRNA#1, #2, and #3 was 38.5%, 76.5%, and 65.2% respectively (Supplementary Fig. 4i).

The reviewer also pointed out that different cell lines should be used to validate the observations. We have indeed used a dozen of colorectal cell lines in this project to support our main conclusion. With regard to Fig. 5 and 6, we verified many of the affected genes in SW480 as well (for example, see Figure 2a-2b in this rebuttal). Most of them showed consistent changes as seen in the HCT116 cells (Fig. 5d-5e). Since the manuscript already contained lots of data, we did not include the verifications in the figures.

For Figure 5H &G, I am confused why need to change to siRNA, and the siHERVH#1/#2 showed low correlation.

The short answer is that, we used siRNA to knockdown different components in the mediator complex, and for better comparison we also synthesized siRNAs for HERVH.

The functional sequences of siRNA are actually identical to the two shRNAs (Table 2). As mentioned above, siRNA#1(shRNA#1) and siRNA#2(shRNA#4) target different subsets of HERVH loci, with the former having a broader spectrum (Table 2). We think this is the main reason for the relatively low correlation between siRNA#1 and #2.

We also reexamined our RNA-seq data with shRNA#1 and shRNA#4, and we found that among the 522 HERVH-dependent upregulated genes identified with shRNA#1 in ARID1A KO HCT116 cells, 427 manifested consistent changes in the cells treated with shRNA#4 (Figure 2c-2d), suggesting the overall phenotypes generated by different HERVH shRNAs are similar.

Figure 2 Experimental validation with different shRNAs and different cell lines.

Overall, the phenomenon reported by the authors are interesting, but more strict quality control will expand my enthusiasm for this paper.

We want to thank the reviewer again for the constructive suggestions, and we hope the reviewer finds our responses satisfactory.